# Mobile barrier mechanisms for Na⁺-coupled symport in an MFS sugar transporter

Parameswaran Hariharan[1], Yuqi Shi[2], Satoshi Katsube[1], Katleen Willibal[3,4], Nathan D Burrows[5], Patrick Mitchell[5], Amirhossein Bakhtiiari[6], Samantha Stanfield[1], Els Pardon[3,4], H Ronald Kaback[7†], Ruibin Liang[6], Jan Steyaert[3,4], Rosa Viner[2], Lan Guan[1]*

[1]Department of Cell Physiology and Molecular Biophysics, Center for Membrane Protein Research, Texas Tech University Health Sciences Center, School of Medicine, Lubbock, United States; [2]Thermo Fisher Scientific, San Jose, United States; [3]VIB-VUB Center for Structural Biology, VIB, Pleinlaan 2, Brussels, Belgium; [4]Structural Biology Brussels, Vrije Universiteit Brussel, Pleinlaan 2, Brussels, Belgium; [5]Division of CryoEM and Bioimaging, Stanford Synchrotron Radiation Light Source, SLAC National Accelerator Laboratory, Menlo Park, United States; [6]Department of Chemistry and Biochemistry, Texas Tech University, Lubbock, United States; [7]Department of Physiology, University of California, Los Angeles, Los Angeles, United States

*For correspondence: lan.guan@ttuhsc.edu

†Deceased

**Abstract** While many 3D structures of cation-coupled transporters have been determined, the mechanistic details governing the obligatory coupling and functional regulations still remain elusive. The bacterial melibiose transporter (MelB) is a prototype of major facilitator superfamily transporters. With a conformation-selective nanobody, we determined a low-sugar affinity inward-facing Na⁺-bound cryoEM structure. The available outward-facing sugar-bound structures showed that the N- and C-terminal residues of the inner barrier contribute to the sugar selectivity. The inward-open conformation shows that the sugar selectivity pocket is also broken when the inner barrier is broken. Isothermal titration calorimetry measurements revealed that this inward-facing conformation trapped by this nanobody exhibited a greatly decreased sugar-binding affinity, suggesting the mechanisms for substrate intracellular release and accumulation. While the inner/outer barrier shift directly regulates the sugar-binding affinity, it has little or no effect on the cation binding, which is supported by molecular dynamics simulations. Furthermore, the hydron/deuterium exchange mass spectrometry analyses allowed us to identify dynamic regions; some regions are involved in the functionally important inner barrier-specific salt-bridge network, which indicates their critical roles in the barrier switching mechanisms for transport. These complementary results provided structural and dynamic insights into the mobile barrier mechanism for cation-coupled symport.

## eLife assessment

In an **important** study that will be of interest to the mechanistic membrane transport community, the authors capture the first cryoEM structure of the inward-facing melibiose transporter MelB, a well-studied model transporter from the major facilitator (MFS) superfamily. CryoEM experiments and supporting biophysical experiments provide **solid** evidence for transporter conformational changes.

## Introduction

Cation-coupled transporters play critical roles in many aspects of cell physiology and pharmacokinetics and are involved in many serious diseases, including cancers, metabolic diseases, neurodegenerative diseases, etc. Their specific locations and functions make them viable drug targets (*César-Razquin et al., 2015*; *Lin et al., 2015*). Following decades of effort, the structures of many transporters have been determined by x-ray crystallography and, more recently, by cryoEM (*Abramson et al., 2003*; *Cherezov et al., 2007*; *Kumar et al., 2014*; *Huang et al., 2003*; *Boudker et al., 2007*; *Yamashita et al., 2005*; *Gouaux, 2009*; *Pedersen et al., 2013*; *Ethayathulla et al., 2014*; *Yan, 2015*; *Drew et al., 2021*; *Guan, 2022*). However, we still do not have a deep understanding of the structural basis for substrate accumulation to high concentrations, the mechanisms for the obligatorily coupled transport, as well as the cellular regulation of transporters' activities to meet metabolic needs.

*Salmonella enterica* serovar Typhimurium melibiose permease (MelB$_{St}$) is the prototype Na$^+$/solute transporter of the major facilitator superfamilies (MFS) (*Ethayathulla et al., 2014*; *Guan, 2018*; *Markham et al., 2021*; *Hariharan and Guan, 2017*; *Hariharan et al., 2018*; *Guan and Hariharan, 2021*; *Blaimschein et al., 2023*; *Granell et al., 2010*; *Ganea et al., 2001*; *Chae et al., 2010*; *Panda et al., 2023*). It catalyzes the stoichiometric symport of galactopyranoside with Na$^+$, H$^+$, or Li$^+$ (*Guan et al., 2011*). Crystal structures of sugar-bound states have been determined, which reveals the sugar recognition mechanism (*Guan and Hariharan, 2021*). In all solved x-ray crystal structures, MelB$_{St}$ is in an outward-facing conformation; the observed physical barrier of the sugar translocation, named the inner barrier, prevents the substrate from moving into the cytoplasm directly. It is well-known that transporters change their conformation to allow the substrate translocation from one side of the membrane to the other and conformational dynamics is the intrinsic feature necessary for their functions (*Abramson et al., 2003*; *Huang et al., 2003*; *Boudker et al., 2007*; *Yan, 2015*; *Drew et al., 2021*; *Guan and Kaback, 2006*; *Khare et al., 2009*).

To understand how the sugar substrate and the coupling cations dictate transporter function, we have established thermodynamic binding cycles in MelB$_{St}$ using isothermal titration calorimetry (ITC) measurements and identified that the positive cooperativity of the binding of cation and melibiose as the core symport mechanism (*Hariharan and Guan, 2017*); furthermore, the binding of the second substrate changes the thermodynamic feature of heat capacity change from positive to negative, suggesting primary dehydration processes when both substrates concurrently occupy the transporter (*Hariharan and Guan, 2021*). While the structural origins have not yet been completely determined yet, they may be related to forming an occluded intermediate conformation, a necessary state for a global conformational transition.

To determine structurally unresolved states of MelB$_{St}$, nanobodies (Nbs) were raised against the wild-type MelB$_{St}$. One group of Nbs (Nb714, Nb725, and Nb733) has been identified to interact with the cytoplasmic side of MelB$_{St}$ and their binding completely inhibits MelB$_{St}$ transport activity (*Katsube et al., 2023*). MelB$_{St}$ bound with Nb725 or Nb733 retained its binding to Na$^+$, but its affinity for melibiose is inhibited. The MelB$_{St}$/Nb733 complex also retained the affinity to its physiological regulator dephosphorylated EIIA$^{Glc}$ of the glucose-specific phosphoenolpyruvate:phosphotransferase system (PTS). EIIA$^{Glc}$, the central metabolite regulator in bacterial catabolic repression (*Postma et al., 1993*; *Deutscher et al., 2014*; *Guan, 2021*), greatly inhibited the galactoside binding to non-PTS permeases MelB$_{St}$ (*Hariharan and Guan, 2021*) and the H$^+$-coupled lactose permease (LacY) (*Hariharan et al., 2015*), but still retained the binding of Na$^+$ to MelB$_{St}$ (*Katsube et al., 2023*). Thus, the functional modulation of the two Nbs on MelB$_{St}$ mimics the binding of EIIA$^{Glc}$. The crystal structure of dephosphorylated EIIA$^{Glc}$ binding to the maltose permease MalFGK$_2$, an ATP-binding cassette transporter, has been determined (*Chen et al., 2013*), but there is no structural information on EIIA$^{Glc}$ binding to the MFS transporters. Determination of the structures of MelB$_{St}$ in complex with this group of Nbs is important and will reveal critical information to understand the manner by which EIIA$^{Glc}$ regulates the large group of MFS sugar transporters.

In the current study, we applied cryoEM single-particle analysis (cryoEM-SPA) and determined the near-atomic resolution structure of MelB$_{St}$ complexed with a Nb725_4, a modified Nb725 with a designed scaffold which enables a NabFab binding (*Bloch et al., 2021*). The use of the Nb provided a Na$^+$-bound lower sugar-affinity inward-facing conformation of MelB$_{St}$. Furthermore, our studies using this conformation-selective Nb in combination with hydrogen/deuterium exchange mass spectrometry (HDX-MS) demonstrate a powerful approach for characterizing protein conformational flexibility

(*Masson et al., 2019*; *Zheng et al., 2019*), and also provides critical dynamic information to understand how the substrate binding drives the conformational transition necessary for the substrate translocation in MelB$_{St}$.

## Results

### CDR grafting to generate a hybrid Nb and its functional characterization

To facilitate structure determination of MelB$_{St}$ complexed with the conformation-selective Nb by cryo-EM-SPA, a novel universal tool based on a NabFab was applied to increase the effective mass of small particles (*Bloch et al., 2021*) since MelB$_{St}$ (mass, 53.5 kDa) has limited extramembrane mass. A hybrid Nb725_4 was created by complementarity-determining region (CDR) grafting or antibody reshaping (*Riechmann et al., 1988*) to transfer the binding specificity of MelB$_{St}$ Nb725 to the TC-Nb4 recognized by the NabFab (*Bloch et al., 2021*; *Figure 1—figure supplement 1*). Extensive in vivo and in vitro functional characterization revealed that the hybrid Nb725_4 possesses the binding properties of its parents Nbs (Nb725 and TC-Nb41) with slightly compromised affinities to MelB$_{St}$ (*Katsube et al., 2023*). Nb725_4 binds to MelB$_{St}$ intracellularly and inhibits MelB-mediated melibiose fermentation and active transport, with an equilibrium dissociation constant ($K_d$) of 3.64 ± 0.62 μM, compared to 1.58 ± 0.42 μM for Nb725 (*Figure 1a–d*; *Figure 1—source data 1*; *Table 1*).

The parent Nb725 had a poor binding to the NabFab (*Figure 1—figure supplement 1*)**,** but Nb725_4 exhibited greatly increased affinity to NabFab at a $K_d$ value of 52.53 ± 13.03 nM (*Figure 1c*, *Table 1*). The anti-Fab Nb binding to NabFab was twofold poorer than Nb725_4 at a $K_d$ value of 112.90 ± 16.33 nM (*Figure 1—figure supplement 1d*, *Table 1*). The complex containing all four proteins, including MelB$_{St}$, Nb725_4, NabFab, and anti-Fab Nb, showed peak-shift in the gel-filtration chromatography (*Figure 1—figure supplement 1e*).

Nb725_4 binding properties were further analyzed in the presence of the MelB ligands. As expected, the presence of melibiose yielded six- and threefold inhibition, and α-NPG afforded eight- and fourfold inhibition, on the binding of Nb725 and Nb725_4, respectively (*Table 1*, *Figure 1—figure supplement 2*). The physiological regulatory protein EIIA$^{Glc}$ (*Table 1*, *Figure 1—figure supplement 2*) showed no inhibition on the binding affinity of both Nbs.

Various ligands binding to the Nb/MelB$_{St}$ complex in the presence of Na$^+$ were also examined with ITC. As published (*Katsube et al., 2023*), the melibiose binding was undetectable in the presence of Nb725 (*Figure 1—figure supplement 3a*). To characterize the sugar effect quantitatively, the α-NPG binding was carried out. Interestingly, the exothermic binding observed in the absence of the Nb changed to endothermic binding reactions in the Nb-bound states; and also, the α-NPG affinity decreased by 21- or 32-fold for Nb725 or Nb725-4, respectively (*Table 2*, *Figure 1—figure supplement 3b*). Inhibitions of galactosides have been observed in the case of EIIA$^{Glc}$/MelB$_{St}$ complex (*Hariharan and Guan, 2014*; *Hariharan et al., 2015*). Collectively, galactosides binding to MelB$_{St}$ were mutually exclusive with Nbs (Nb725, Nb725_4, or Nb733) or EIIA$^{Glc}$, and the negative cooperativity implied that MelB$_{St}$ conformations favored by galactosides and those binders differ. This conclusion was further supported by the Na$^+$ or EIIA$^{Glc}$ binding to the MelB$_{St}$/Nb complexes (*Table 2*, *Figure 1—figure supplement 3c and d*), which showed no inhibition for Na$^+$ affinity and even slightly better binding for EIIA$^{Glc}$. Therefore, MelB$_{St}$ complexed with Nb725_4 retains its physiological functions, and its conformation is expected to be similar when trapped by either of those Nbs and EIIA$^{Glc}$ but is different from the sugar-bound outward-facing structure (PDB ID 7L17) (*Guan and Hariharan, 2021*).

### Inward-facing MelB$_{St}$ trapped by Nb725_4

CryoEM-SPA was used to image the complex consisting of MelB$_{St}$ in nanodiscs, Nb725_4, NabFab, and anti-Fab Nb. The data collection, processing, and evaluation followed the standard protocols (*Figure 2—figure supplements 1–4*). From a total number of 296,925 selected particles, structures of MelB$_{St}$, Nb725_4, and NabFab were determined to a golden standard Fourier shell correlation (GSFSC) resolution of 3.29 Å at 0.143 (*Figure 2*). The map for anti-Fab Nb was relatively poor and excluded from particle reconstruction. The quality of the Coulomb potential map allowed for unambiguous docking of available coordinates of NabFab (PDB ID 7PHP), a predicted model for Nb725_4 by Alpha-Fold 2 (*Jumper et al., 2021*; *Bryant et al., 2022*), and the N-terminal helix bundle from the

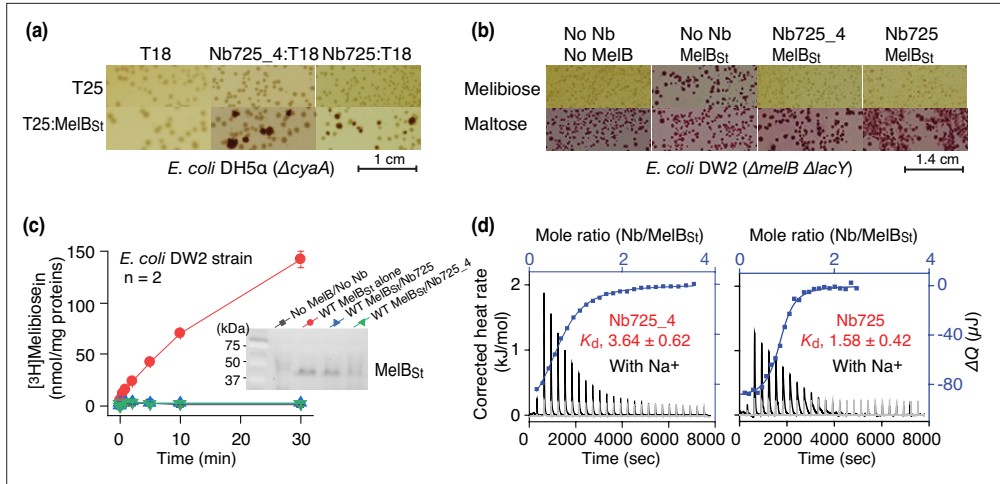

**Figure 1.** Functional characterizations of the hybrid Nb725_4. (**a**) In vivo two-hybrid interaction assay. Two compatible plasmids encoding T25:MelB$_{St}$ and Nb:T18 were transformed into *E. coli* DH5α cyaA cells and plated on the maltose-containing MacConkey agar plate as described in 'Materials and methods'. The irregular red colonies are typical of a positive two-hybrid test indicating the protein-protein interactions. The image for the two hybrids T25:MelB$_{St}$ and Nb725:T18 was reused from Figure 3a in *Katsube et al., 2023* under a Creative Commons Attribution (CC BY 4.0) license. (**b**) Inhibition of melibiose fermentation by Nbs. Two compatible plasmids encoding MelB$_{St}$ and Nb725 or Nb725_4 were transformed into *E. coli* DW2 cells [ΔmelBΔlacYZ] and plated on the MacConkey agar plate containing maltose (as the positive control) and melibiose (for testing transport activity of MelB$_{St}$) as the sole carbon source. The image of MelB$_{St}$/Nb725 was reused (*Katsube et al., 2023*). (**c**) [³H] Melibiose transport assay with *E. coli* DW2 cells. The cells transformed by two compatible plasmids encoding the MelB$_{St}$ and Nb725 or Nb725_4 were prepared for [³H]melibiose transport assay at 0.4 mM (specific activity of 10 mCi/mmol) and 20 mM Na⁺ as described in 'Materials and methods'. The cells transformed with the two empty plasmids without MelB or Nb were the negative control. Inset, western blot. MelB$_{St}$ expression under the co-expression system was analyzed by isolating the membrane fractions. An aliquot of 50 μg was loaded on each well and MelB$_{St}$ protein was probed by the HisProbe-HRP Conjugate. (**d**) Nb binding to MelB$_{St}$ by isothermal titration calorimetry (ITC) measurements. As described in 'Materials and methods', the thermograms were collected with the Nano-ITC device (TA Instrument) at 25°C. Exothermic thermograms shown as positive peaks were obtained by titrating Nbs (0.3 mM) into the MelB$_{St}$-free buffer (gray) or MelB$_{St}$ (35 μM)-containing buffer (black) in the Sample Cell and plotted using bottom/left (x/y) axes. The binding isotherm and fitting of the mole ratio (Nb/MelB$_{St}$) vs. the total heat change (ΔQ) using one-site independent-binding model were presented by top/right (x/y) axes. The dissociation constant $K_d$ was presented at mean ± SEM (number of tests = 6–7).

The online version of this article includes the following source data and figure supplement(s) for figure 1:

**Source data 1.** Original photographs that were taken from in vivo two-hybrid interaction assay and used to prepare *Figure 1a*.

**Source data 2.** Original photographs that were taken from sugar fermentation assay results and used to prepare *Figure 1b*.

**Source data 3.** Western blot to detect MelB$_{St}$ expression when co-expressing with Nb725 or Nb725_4 (unlabelled).

**Source data 4.** Western blot to detect MelB$_{St}$ expression when co-expressing with Nb725 or Nb725_4 (labelled).

**Figure supplement 1.** Hybrid Nb725_4 generated by complementarity-determining region (CDR) grafting.

**Figure supplement 1—source data 1.** Western blot shown in *Figure 1—figure supplement 1e* (unlabelled).

**Figure supplement 1—source data 2.** Western blot shown in *Figure 1—figure supplement 1e* (labelled).

**Figure supplement 2.** Ligand effects on the nanobodies (Nbs) binding.

**Figure supplement 3.** Nanobody (Nb) effects on substrate/ligand binding.

---

outward-facing crystal structure of D59C MelB$_{St}$ (PDB ID 7L17). The C-terminal helix bundle of MelB$_{St}$ and the ion Na⁺ were manually built. The final model contains 417 MelB$_{St}$ residues (positions 2–210, 219–355, and 362–432), with 6 unassigned side chains at the C-terminal domain (Leu293, Tyr355, Arg363, Tyr369, Tyr396, and Met410) due to the map disorder, 122 of Nb725_4 residues 2–123, and 229 of NabFab H-chain residues 1–214, and 210 of NabFab L-chain residues 4–213, respectively. The

**Table 1.** Nanobodies (Nbs) binding.

| Titrant | Titrate condition | $K_d$ (μM) | Number of tests | Fold of affinity change | p-Value* |
|---|---|---|---|---|---|
| | MelB$_{St}$/Na$^+$ | 3.64 ± 0.62† | 7 | | |
| | MelB$_{St}$/Na$^+$/melibiose | 11.81 ± 0.72 | 6 | -3.24 | <0.01 |
| | MelB$_{St}$/Na$^+$/α-†NPG | 14.63 ± 1.27 | 2 | -4.02 | <0.01 |
| | MelB$_{St}$/Na$^+$/EIIA$^{Glc}$ | 2.14 ± 0.11 | 2 | +1.70 | >0.05 |
| Nb725_4 | NabFab | 52.53 ± 13.03‡ | 2 | | |
| | MelB$_{St}$/Na$^+$ | 1.58 ± 0.42 | 6 | | |
| | MelB$_{St}$/Na$^+$/melibiose | 8.88 ± 1.24 | 5 | -5.62 | <0.01 |
| | MelB$_{St}$/Na$^+$/α-NPG | 12.58 ± 0.60 | 2 | -7.96 | <0.01 |
| | MelB$_{St}$/Na$^+$/EIIA$^{Glc}$ | 1.18 ± 0.12 | 2 | +1.34 | >0.05 |
| Nb725 | NabFab | 36.02 | 1 | | |
| Anti-Fab Nb | NabFab | 112.90 ± 16.33‡ | 2 | | |

*Unpaired *t*-test.
†Mean ± SEM.
‡nM.

local resolution estimate showed that the cores of NabFab, Nb725_4, and the N-terminal helix bundle of MelB$_{St}$ exhibit better resolutions up to 2.84 Å and most regions in the C-terminal helix bundle of MelB$_{St}$ have resolutions between 3.2 and 3.6 Å (*Figure 2—figure supplement 5a*). The N-terminal helix bundle has a completely connected density; however, the map corresponding to three

**Table 2.** Nanobody (Nb) effects on MelB$_{St}$ binding to sugar, Na$^+$, and EIIA$^{Glc}$.

| Titrant | Titrate condition | $K_d$ (μM) | Number of tests | Fold of affinity change | p-Value* | Reference |
|---|---|---|---|---|---|---|
| | MelB$_{St}$/Na$^+$ | 1430 ± 30.0† | 5 | / | | |
| | MelB$_{St}$/Na$^+$/Nb725_4 | /‡ | 3 | | | |
| | MelB$_{St}$/Na$^+$/Nb725 | / | 3 | | | |
| Melibiose | MelB$_{St}$/Na$^+$/EIIA$^{Glc}$ | / | 3 | | | *Hariharan et al., 2015* |
| | MelB$_{St}$/Na$^+$ | 16.46 ± 0.21 | 5 | / | | |
| | MelB$_{St}$/Na$^+$/Nb725_4 | 531.55 ± 19.25 | 2 | -32.29 | <0.01 | |
| | MelB$_{St}$/Na$^+$/Nb725 | 353.55 ± 37.25 | 2 | -21.48 | <0.01 | |
| α-NPG | MelB$_{St}$/Na$^+$/EIIA$^{Glc}$ | 76.13 ± 4.52 | 2 | -4.63 | <0.01 | *Hariharan et al., 2015* |
| | MelB$_{St}$ | 261.77 ± 49.15 | 4 | / | | |
| | MelB$_{St}$/Nb725_4 | 345.04 ± 9.92 | 2 | -1.32 | >0.05 | |
| | MelB$_{St}$/Nb725 | 182.50 ± 11.30 | 2 | +1.43 | >0.05 | |
| Na$^+$ | MelB$_{St}$/EIIA$^{Glc}$ | 253.50 ± 11.00 | 2 | +1.03 | >0.05 | *Katsube et al., 2023* |
| | MelB$_{St}$/Na$^+$ | 3.76 ± 0.29 | 5 | / | | |
| | MelB$_{St}$/Na$^+$/Nb725_4 | 4.32 ± 0.64 | 4 | -1.15 | >0.05 | |
| EIIA$^{Glc}$ | MelB$_{St}$/Na$^+$/Nb725 | 1.94 ± 0.10 | 4 | +1.94 | <0.01 | |

*Unpaired *t*-test.
†Mean ± SEM.
‡Not detectable.

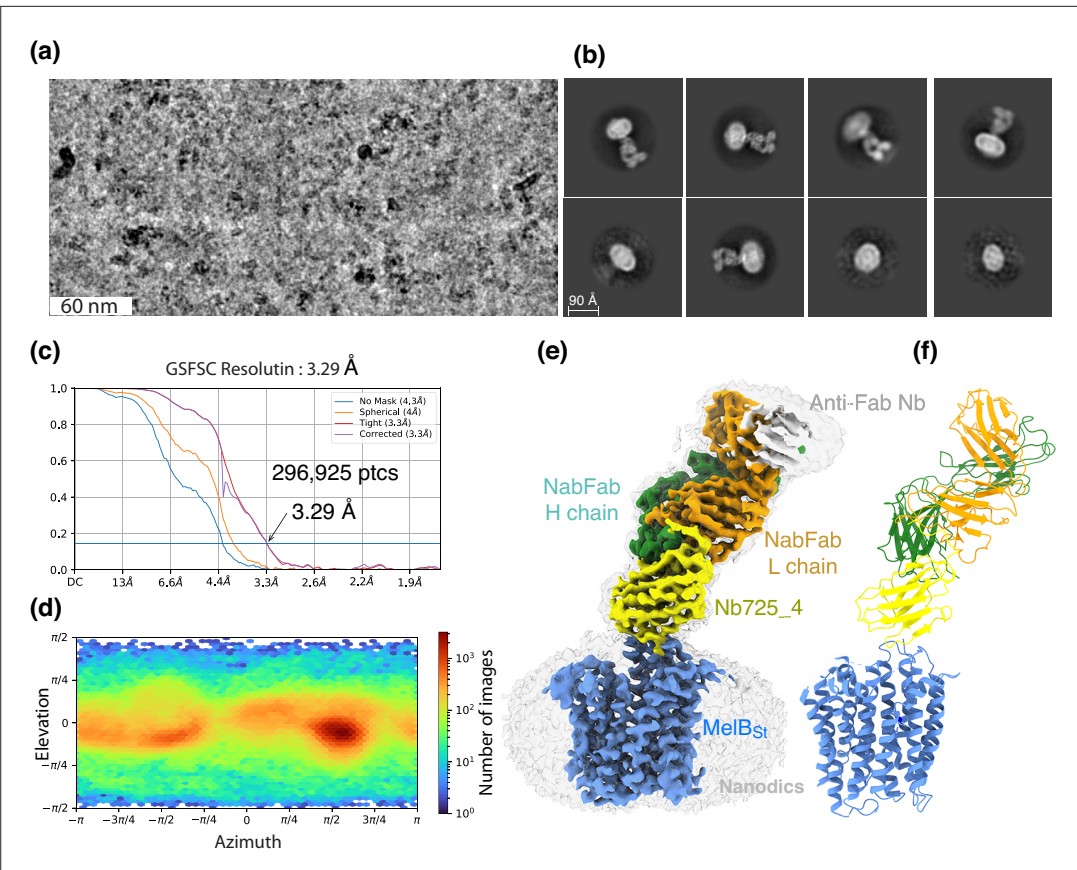

**Figure 2.** CryoEM single-particle analysis (cryoEM-SPA). The samplecoum containing the wild-type MelB$_{St}$ in lipids nanodiscs, the MelB$_{St}$-specific Nb725_4, NabFab, and anti-Fab Nb at 1.5 mg/mL in 20 mM Tris–HCl, pH 7.5, and 150 mM NaCl was prepared as described in 'Materials and methods'. Images were collected using Titan Krios TEM with a K3 detector of S$^2$C$^2$, Stanford, CA. The particle reconstructions and modeling were performed as described in 'Materials and methods'. The final volume did not include the anti-Fab Nb during Local Refinement due to relatively poor densities. (**a**) The raw image after motion correction. (**b**) Representative 2D classes generated by CryoSPARC program. MelB$_{St}$ in nanodiscs, Nb725_4, NabFab, and anti-Fab Nb can be easily recognized. (**c**) Golden standard Fourier shell correlation (GSFSC) resolution was calculated by cryoSPARC Validation (FSC) using two half maps generated by the CryoSPARC Local Refinement program. The number of particles used for the volume reconstruction is presented. (**d**) Particle distribution of orientations over azimuth and elevation angles generated by CryoSPARC Local Refinement program. (**e, f**) The structure of MelB$_{St}$/Nb725_4/NaFab complex. The volume (**e**) and cartoon representation (**f**) were colored by polypeptide chains as indicated. Nanodiscs were transparent and colored in light gray. Sphere and sticks in panel (**f**) highlighted Na$^+$ and its ligands.

The online version of this article includes the following source data and figure supplement(s) for figure 2:

**Figure supplement 1.** CryoEM data process.

**Figure supplement 2.** Initial 3D reconstruction.

**Figure supplement 3.** Refinement and map improvement.

**Figure supplement 4.** Golden standard Fourier shell correlation (GSFSC) resolution and 3dFSC.

**Figure supplement 5.** Evaluation of map and models.

**Figure supplement 6.** NabFab comparison.

**Figure supplement 7.** Interactions of Nb725m_4 and MelB$_{St}$.

**Figure supplement 8.** Complex of MelB$_{St}$ with Nb725 and EIIA$^{Glc}$.

**Figure supplement 8—source data 1.** Western blot shown in *Figure 2—figure supplement 8* (unlabelled).

**Figure supplement 8—source data 2.** Western blot shown in *Figure 2—figure supplement 8* (labelled).

cytoplasmic loops at positions 211–218 in the middle loop$_{6-7}$, positions 356–361 in loop$_{10-11}$, or the entire C-terminal tail after Tyr432, is missing. The map quality, model statistics, and the model-map matching Q scores generally matched to this reporting resolution (*Table 3*, *Figure 2—figure supplement 5a–c*).

The NabFab structure in this complex is virtually identical to that in the NorM complex (PDB ID 7PHP) (*Bloch et al., 2021*) with rmsd of all atoms of 1.334 Å (*Figure 2—figure supplement 6a–c*). When the alignment focused on the NabFab H-chain, the two NabFab/Nb binary complexes in both complexes were organized similarly (*Figure 2—figure supplement 6d*), and the binding of Nbs with their corresponding transporters varied.

The structure shows that the Nb725_4 is bound to the cytoplasmic side of MelB$_{St}$ at an inward-open conformation (*Figure 2—figure supplement 7a–c*), which supports the functional analyses (*Katsube et al., 2023*) and reveals the MelB$_{St}$ conformation for the Nb binding. The contact surfaces match in shape and surface potential with a contact area of 435 A$^2$. CDR-1 (Arg30, Asp31, Asn32, and Ala33) and CDR-2 (Tyr52, Asp53, Leu54, Tyr56, Thr57, and Ala58) form interactions with two cytoplasmic regions of MelB$_{St}$, the tip between helices IV and V, that is, loop$_{4-5}$ (Pro132, Thr133, Asp137, Lys138, and Arg141) and the beginning region of the middle loop$_{6-7}$ (Tyr205, Ser206, and Ser207) (*Figure 2—figure supplement 7c*). The Nb binding was stabilized by salt-bridge and hydrogen-bonding interactions at both corners of this stretch of inter-protein contacts. In the published outward-open conformation of MelB$_{St}$, such a binding surface for recognizing Nb725_4 does not exist. Most regions of the middle loop, the C-terminal helix bundle, and the C-terminal tail helix must be displaced to permit the binding, and some of those regions are completely disordered in this inward-facing structure. All data supported that Nb725_4 only binds to MelB$_{St}$ in an inward-open state. Thus, it is a conformation-selective Nb.

Nb725_4, Nb725, and another Nb733 (*Katsube et al., 2023*) behave similarly; all inhibit the sugar binding and transport. We conclude that all three binders are the inward-facing conformation-specific negative modulators of MelB$_{St}$. Further, ITC measurements (*Figure 1—figure supplement 2*) showed that MelB$_{St}$ binds to either of three Nbs in the absence or presence of EIIA$^{Glc}$ or binds to EIIA$^{Glc}$ in the absence or presence of either Nbs. The complex composed of MelB$_{St}$, EIIA$^{Glc}$, and Nb725 (or Nb733; *Katsube et al., 2023*) can be isolated (*Figure 2—figure supplement 8*).

## MelB$_{St}$ trapped in a Na$^+$-bound, low sugar-affinity state

Between helices II and IV (*Figure 3a and b*), one Na$^+$ cation was modeled, liganded by four side-chains, Asp55, Asn58, Asp59 (helix II), and Thr121 (helix IV), at distances of between 2.1 and 2.8 Å. The backbone carbonyl oxygen of Asp55 is also in close proximity to the bound Na$^+$. The local resolution of this binding pocket is approximately 2.9–3.2 Å. The model to map Q-scores for Na$^+$ and the binding residues and side chains are generally greater than the expected scores at this resolution except for the Thr121 side chain, which has a lower score (*Pintilie et al., 2020*). The observation of this cation-binding pocket is supported by previous extensive data (*Hariharan and Guan, 2017*; *Ding and Wilson, 2001*; *Katsube et al., 2022*). Cys replacement at position Asp59 changes the symporter to a uniporter (*Guan and Hariharan, 2021*). The D59C mutant does not bind to Na$^+$ or Li$^+$ as well as loses all three modes of cation-coupled transport but catalyzes melibiose fermentation in cells (melibiose concentration-driven transport) (*Guan and Hariharan, 2021*) or melibiose exchange in membrane vesicles (*Ethayathulla et al., 2014*). Asp55 also plays a critical role in the binding of Na$^+$ or Li$^+$ and in the Na$^+$- or Li$^+$-coupled transport (*Ethayathulla et al., 2014*; *Hariharan and Guan, 2017*). Extensive studies of *Escherichia coli* MelB also show that both negatively charged residues are involved in cation binding (*Granell et al., 2010*; *Pourcher et al., 1993*; *Zani et al., 1993*). Thr121 has been determined to be critical for Na$^+$, not H$^+$ nor Li$^+$ (*Katsube et al., 2022*); and the Cys mutation at Asn58 also loses Na$^+$ binding (*Ethayathulla et al., 2014*). Interestingly, the Na$^+$ recognition can be selectively eliminated by specific mutations. Further, the Na$^+$ binding is supported by MD simulations at both inward and outward states using MD simulations in the absence of the Nb (*Figure 3—figure supplement 1*), which reveals the ideal coordination geometry for Na$^+$ binding.

The crystal structure of sugar-bound D59C MelB$_{St}$ (*Figure 3c and d*, *Figure 3—figure supplement 2*), which retains an intact sugar-binding and translocation pathway, shows that the sugar-binding pocket is formed by both N- and C-terminal bundles (*Guan and Hariharan, 2021*). Due to the Cys mutation, the cation-binding pocket is loosely packed and this mutant does not bind Na$^+$ or Li$^+$. When

**Table 3.** CryoEM data collection and structure determination statistics.

| | MelB$_{st}$/Nb725m/NabFab | |
|---|---|---|
| EMDB | EMD-41062 | |
| PDB | 8T60 | |
| | **Non-tilted collection** | **Tilted collection** |
| **Data collection** | | |
| Microscope | Krios-TEMBETA | Krios-TEMBETA |
| Voltage (kV) | 300 | 300 |
| Number of movies | 14,094 | 8716 |
| Electron dose (e$^-$/Å$^2$) | 50.00 | 50.00 |
| Defocus range (μm) | −0.8 to −1.8 | −0.8 to −1.8 |
| Pixel size (Å) | 0.86 | 0.86 |
| Plate tilt angel (°) | / | 30 |
| **Data processing** | | |
| Initial number of particles | 7,632,727 | 2,887,147 |
| Combined final number of particles | 203,876 | |
| Symmetry imposed | C1 | |
| Map resolution* (Å) | 3.29 | |
| B factor | 101.4 | |
| **Model refinement** | | |
| Chains | 5 | |
| Non-hydrogen atoms | 7503 | |
| Protein residues | 978 | |
| **Mean B-factor** | | |
| Protein | 97.35 | |
| Na$^+$ ion | 88.64 | |
| **RMS deviations** | | |
| Bond lengths (Å) | 0.003 | |
| Bond angles (°) | 0.509 | |
| MolProbity score | 1.82 | |
| Clash score | 4.09 | |
| Poor rotamers (%) | 1.98 | |
| **Ramachandran plot** | | |
| Favored (%) | 93.48 | |
| Allowed (%) | 6.52 | |
| Outliers (%) | 0.00 | |
| Model resolution (Å)† | 3.2/3.3/3.5 | |

*Resolution determined by Fourier shell coefficient threshold of 0.143 for corrected masked map.
†Resolution determined between the model and the resolved map by Fourier shell coefficient threshold of 0/0.143/0.5.

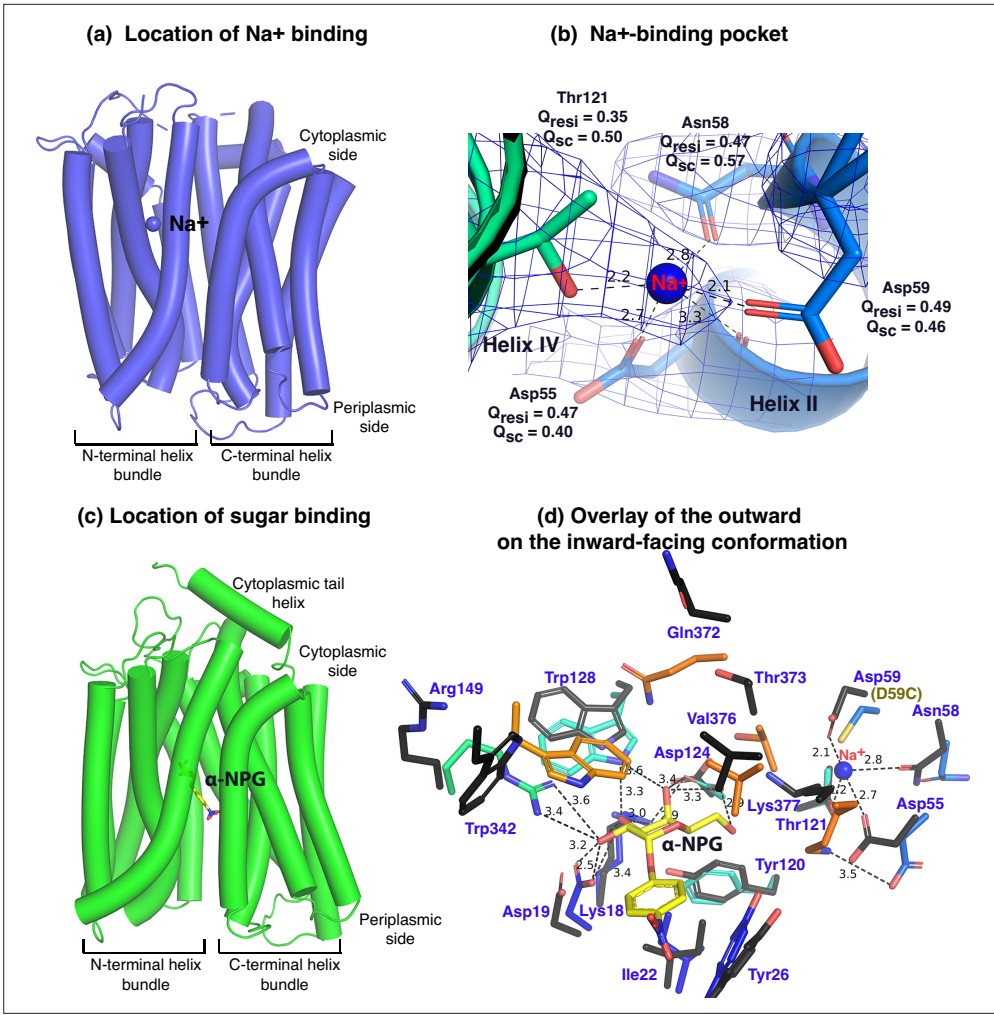

**Figure 3.** Na⁺- and sugar-binding pockets of MelB$_{St}$. (**a**) Location of the Na⁺ binding site. The inward-facing cryoEM structure of the WT MelB$_{St}$ is displayed in cylindrical helices with the cytoplasmic side on the top. One bound Na⁺ ion within the N-terminal helix bundle was shown in the blue sphere. (**b**) Na⁺-binding pocket. The isomesh map of the inward-facing conformation was created by the Pymol program using level 10 and carve of 1.8. The Na⁺ coordinates are shown in dashed lines (Å) and interacting residues are shown in sticks. $Q_{res}$, Q score for residue; $Q_{sc}$, Q score for side chain. (**c**) Location of the galactoside-binding stie. The outward-facing x-ray crystal structure of D59C MelB$_{St}$ mutant is displayed in cylindrical helices with the cytoplasmic side on the top (PDB ID 7L17). One α-nitrophenyl galactoside (α-NPG) molecule is shown in the stick colored in yellow between the N- and C-terminal helix bundles. (**d**) Superimposed sugar- and cation-binding pockets. The α-NPG-bound outward-facing structure in c was aligned with the inward-facing cryoEM structure based on the 2–200 region. The residues in the sugar- and cation-binding pockets of the inward-facing cryoEM structure are colored in black and labeled in blue. D59C of the α-NPG-bound outward-facing structure is indicated in the parentheses. The α-NPG and Na⁺ are colored yellow and blue, respectively.

The online version of this article includes the following figure supplement(s) for figure 3:

**Figure supplement 1.** MD simulations of the Na⁺ binding at both inward- and outward-facing states.

**Figure supplement 2.** Galactose-binding pocket in the outward-facing crystal structure (PDB ID 7L17).

**Figure supplement 3.** The alignment of the sugar-bound outward-facing structure (7L17, green) with the Na+ -bound inward-facing structure (blue) was carried out in Pymol or Coot programs.

**Figure supplement 4.** Membrane topology.

overlaying the outward-facing crystal structure upon the inward-facing cryoEM structures (*Figure 3d*, sticks in black), little variation of the positioning of the sugar-binding residues on the N-terminal bundle is observed except for Arg149, but all C-terminal residues involved in the sugar-binding pocket show greater displacements. In the cation-site mutant, Lys377 forms a salt-bridge interaction with Asp55; in the $Na^+$-bound structure, Lys377 is closer to Asp124, and Asp55 is liganded with $Na^+$.

The loosely arranged sugar-binding residues observed in the Nb725_4-bound conformation provide the structural foundation for measured lower sugar affinities (*Table 2*), so this structure represents an intracellular $Na^+$-bound sugar-releasing conformation. Since the positioning of the key residues for galactoside specificity (Lys18, Asp19, and Asp124) exhibits virtually no change, it is conceivable that this conformation retains the initial recognition of sugar for reversal reactions, a common feature for carrier transporters (*Guan et al., 2011*; *Guan and Kaback, 2009*).

## Conformational changes between inward- and outward-facing states

The outward-facing structure was trimmed to match the residues resolved in the inward structure. The superposition of the outward- and inward-facing $MelB_{St}$ exhibited rmsd of all atoms of 5.035 Å (*Figure 3—figure supplement 3*), and the major displacement exists at both ends of transmembrane helices. The alignment based on the N-terminal positions 2–200 or the C-terminal positions 231–432 exhibits a clear difference in the angle of the domain packing (*Figure 3—figure supplement 3b and c*). The focused alignment of positions 2–200 or 231–432 exhibits greatly reduced rmsd values (*Figure 3—figure supplement 3d and e*). The N-terminal domains at both conformations are virtually the same, especially for the helices I, II, and IV that host the sugar- and cation-specificity determinant pockets (*Figure 3—figure supplement 3f*). Some structural rearrangements within 3–5 Å exist in the C-terminal bundle, especially on both ends of cavity-lining helices (VII, VIII, X, and XI) and their link or extended loop (*Figure 3—figure supplement 3g and h*). Thus, the conformational changes between the outward- and inward-facing states can be adequately described by a rocker-switch-like movement with some structural rearrangements in the C-terminal bundle as also indicated in *Figure 3—figure supplement 4*.

## Substrate translocation barriers

The two structures clearly reveal the two essential barriers critical for substrate translocation, the thicker inner barrier in the outward structure and the thinner outer barrier in the inward structure (*Figure 4a and d*). The core of the inner barrier is formed by three-helix pairs from both domains (V/VIII, IV/X, and back II/XI), and stabilized by an extensive charged network (*Figure 4b*; *Ethayathulla et al., 2014*; *Markham et al., 2021*; *Guan and Hariharan, 2021*; *Amin et al., 2014*). The N-terminal Arg141 (helix V) forms salt-bridge interactions with the C-terminal Asp351 and Asp354 (helix X), and this core interaction links with four other charged residues to stabilize the cytoplasmic closure by forming an inner barrier. In addition, the inner barrier is also stabilized by two peripheral amphiphilic helices at the cytoplasmic side (in the middle loop and the C-terminal tail) (*Figure 4b*). The entire middle loop$_{6-7}$ across the two domains interact with broader areas (the N-terminal loop$_{2-3}$, loop$_{4-5}$, and helices II and IV, as well as the C-terminal loop$_{10-11}$, helices X and XI, and the C-terminal tail). The C-terminal tail helix interacts with both N- and C-terminal domains (helices V and X). The contacting area between the two bundles (based on positions 2–212 and 213–450) is 2218 Å (*Lin et al., 2015*) with nine salt-bridge and five hydrogen-bounding interactions.

The core of the outer barrier in the inward-facing structure is also formed by three pairs of helices from both domains (V/VIII, I/VII, and II/XI) (*Figure 4f*). The longest loop$_{11-12}$ at the periplasmic side moves into the middle area and forms contacts with helices I and VII, sealing the periplasmic opening. Notably, helical pair I/VII or IV/X is only involved in the outer or inner barrier, respectively, and the other two cavity-lining helical pairs (V/VIII and II/XI) are engaged in both barriers (*Figure 4b and c, e, f*). The contact area between the two bundles at the periplasmic side of 1435 Å (*Lin et al., 2015*) is much smaller. These structural features further supported that the outward-facing state is more stable than the inward-facing $MelB_{St}$. In the inward-facing conformation, all of the salt-bridge interactions are broken (*Figure 4b and e*), and the Arg141 and Lys138 engage in the binding of Nb725_4 (*Figure 2—figure supplement 7*). Therefore, this cytoplasmic-side salt-bridge network is the inner barrier-specific.

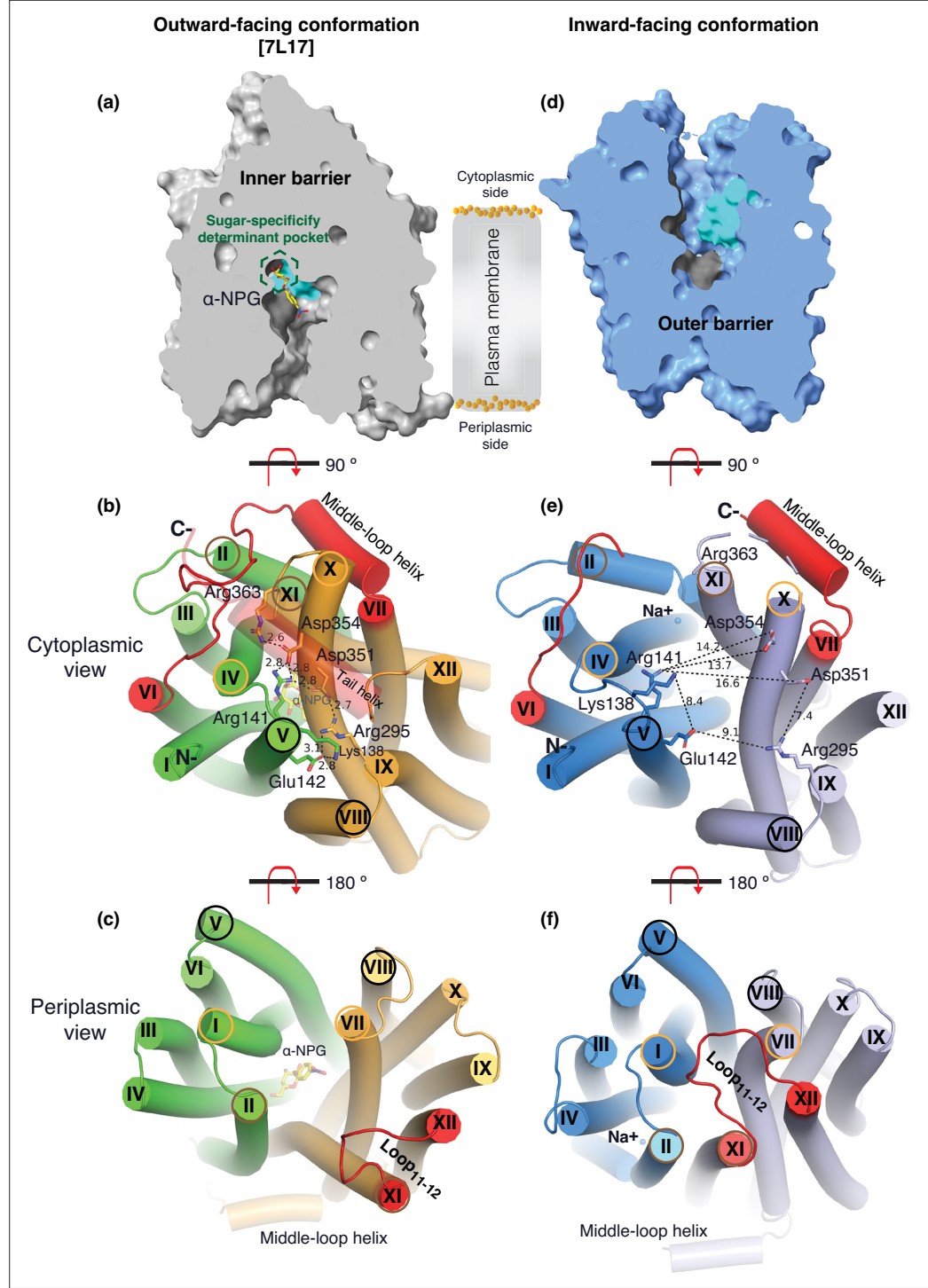

**Figure 4.** Barriers and sugar-binding pocket. Outward-facing (PDB ID 7L17; left column) and inward-facing (right column) structures were used to prepare the figures. (**a, d**) Side view with cytoplasmic side on top. The inner and outer barriers are labeled. The sugar-specificity determinant pocket is highlighted in a dashed hexagon. The residues contributing to the bound α-NPG in N- and C-terminal bundles are colored in dark gray and cyan, respectively. (**b, e**) Cytoplasmic view. The charged network between the N- and C-terminal bundles is colored in green and bright orange in panel (**b**), or blue and light blue in panel (**e**), respectively. The C-terminal tail helix was set in transparent in panel (**b**) but disordered in the inward-facing conformation in panel (**e**). The charged residues are highlighted in sticks. Arg363 side chain missed the side chain in the inward-facing structure. (**c, f**) Periplasmic view. The paired helices involved in either barrier formation are highlighted in the same colored circles. The α-NPG

*Figure 4 continued on next page*

*Figure 4 continued*

is colored in yellow and Na$^+$ is shown in blue sphere. The cytoplasmic middle loop, the C-terminal tail, and the periplasmic loop$_{11-12}$ are highlighted in red. Distance between two residues is shown by dashed lines (Å).

Assessed by the sugar-bound structure (*Figure 4a*), the substrate-binding pocket is outlined by Trp128 and Trp 342 on the inner barrier and Tyr26 on the outer barrier (*Figure 3d*, *Figure 3—figure supplement 1*), part of both barriers. Notably, the major binding residues that involve the multiple hydrogen-bonding interactions with the galactosyl moiety are only part of the inner barrier. In the inward-facing state, the inner barrier is broken, which exposes the binding site to the cytoplasm and also results in the interruption of the sugar-binding pocket (*Figure 4d*). The C-terminal binding residues moved away due to the displacement of the backbone of helices X/XI, and the sugar-binding residues become loosely organized, resulting in a low sugar-binding affinity. The structural features indicated that the molecular mechanisms for the sugar translocation simply result from the barrier switch generating a low sugar-affinity inward state of MelB. The outer barrier does not contain residues essential for galactoside binding, so the break of the outer barrier has a much less profound effect on the sugar-binding affinity.

## Conformational dynamics measured by HDX-MS

Bottom-up HDX-MS measures the exchange rate of amide hydrogens with deuterium by determination of time-dependent peptide level mass spectra (*Masson et al., 2019*). It is a label-free quantitative technique that can disclose the dynamics of a full-length protein simultaneously. This approach was utilized to determine MelB$_{St}$ dynamics in this study. The Na$^+$-bound WT MelB$_{St}$ proteins exist in multiple conformations including at least the outward- and inward-facing states but strongly favor the outward-facing states. The Nb725_4-trapped MelB$_{St}$ exists only in an inward-facing conformation. A pair of WT MelB$_{St}$ and WT MelB$_{St}$ complexed with Nb725_4 was subjected to in-solution HDX-MS experiments. MelB$_{St}$ yields 86% coverage with 155 overlapping peptides at an average peptide length of 9.3 residues (*Figure 5a and b*, *Figure 5—source data 1*, *Table 4*), leaving 66 non-covered residues; most are located in the middle regions of transmembrane helices or in a few cytoplasmic loops (*Figure 6*, ball in gray).

The effect of MelB$_{St}$ structural dynamics imposed by Nb725_4 is presented as the differential deuterium labeling (ΔD). As shown in the residual plot of each peptide between the two states (D$_{Nb725\_4\text{-bound} - Nb\text{-free}}$) (*Figure 5b*) at three labeling time points, variable levels of protections (less deuterium exchange in the complex state) are observed in MelB$_{St}$. A slight deprotection is only observed at the N-terminus. After proper statistical analysis, there are 232 positions (>50% labeled positions) exhibiting insignificant HDX changes (D$_{Nb725\_4\text{-bound} - Nb\text{-free}}$ value < |0.3184| or p>0.05) as indicated by ribbon representation (*Figure 5b*). The remaining 177 positions, not counting the His tag, covered by 71 overlapping peptides show statistically significant changes (D$_{Nb725\_4\text{-bound} - Nb\text{-free}}$ value ≥ |0.3184| and p≤0.05) and are highlighted by the cartoon representation (*Figure 6*). Overall, those regions are mainly distributed on the cavity-lining helices and cytoplasmic peripheral regions including the middle loop and C-terminal tails, as well as the C-terminal periplasmic loops. The corresponding deuterium uptake curves covering three-magnitude time points were plotted (*Figure 6—figure supplements 1 and 2*) and the representative peptides are displayed as insets around the structure (*Figure 6*, insets). Most Nb725_4 binding residues are not covered, either not covered by the labeled peptides (Tyr205, Ser206, and Ser207) or by insignificant change (Pro132 and Thr133), except for Asp137, Lys138, and Arg141 that were presented in eight peptides. Depending upon the exchange rate and response to the binding of Nb725_4, the MelB$_{St}$ HDX can be categorized into four groups.

Group I peptides show low-level HDX exchange rates and Nb725_4 afforded complete inhibition for up to at least 50 min (*Figure 6*, cartoon and curve in blue). These peptides are located in transmembrane helices, middle-loop helix, and C-terminal periplasmic loops across both domains and most are clustered at the cavity-lining helices. Peptides carrying the Nb-binding residues Asp137, Lys138, and Arg141 are marked in the plots (*Figure 6*, inset; *Figure 6—figure supplements 1 and 2*). Group II peptides show faster deuterium uptake over time (*Figure 6*, cartoon and curve in deep olive). The Nb725_4 binding affords small to moderate protections. The group II peptides are spotted at three cytoplasmic regions; interestingly, all surround the helix X, which plays a critical role in the stability of the inner barrier. At the inward-facing structure, most of those residues are either missing

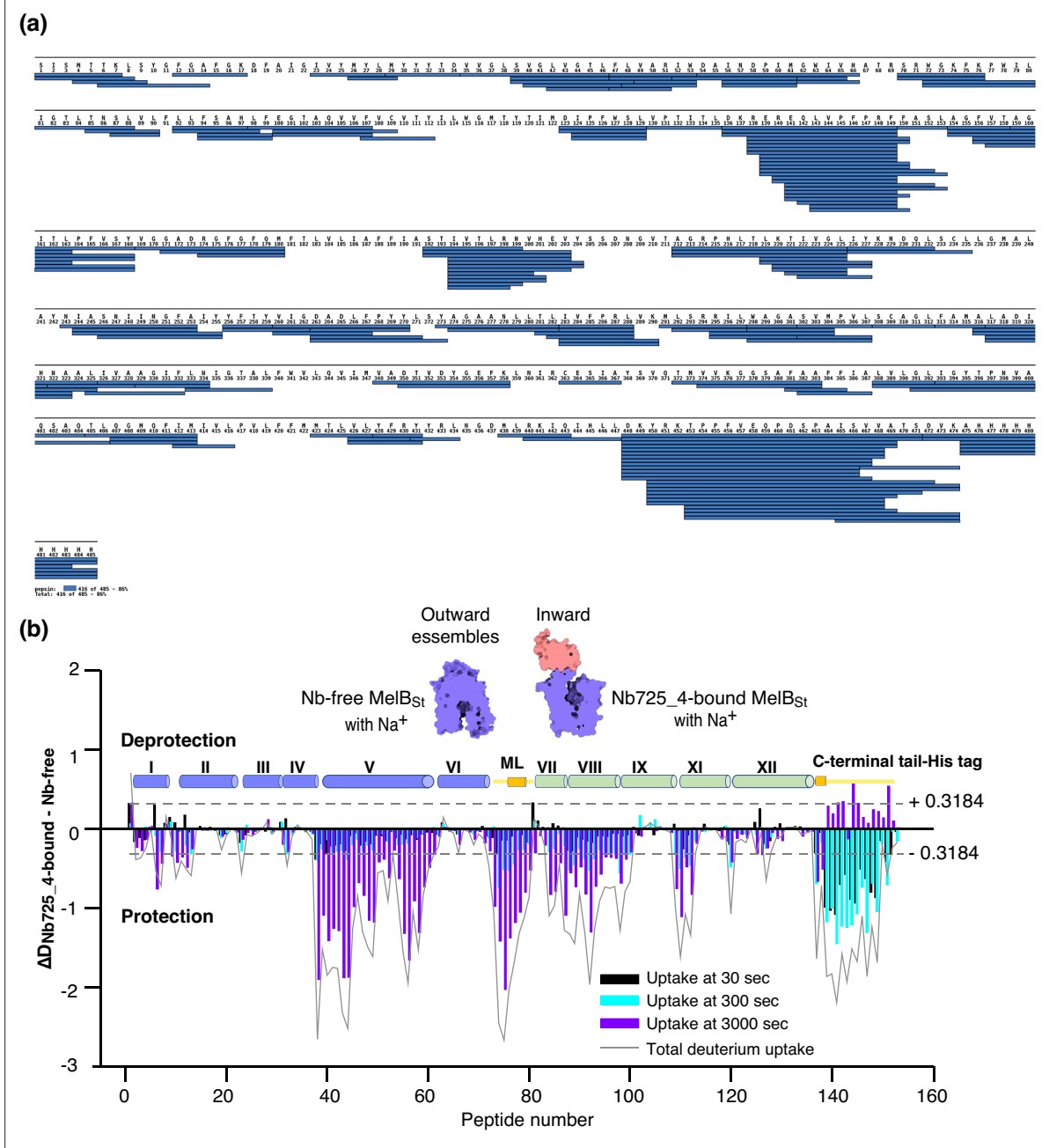

**Figure 5.** MelB_St dynamics probed by hydrogen/deuterium exchange mass spectrometry (HDX-MS). (**a**) MelB_St peptide sequence coverage. The peptides of the deuterated MelB_St were determined based on the MelB_St peptide database that was generated by nonspecific digestions of non-deuterated MelB_St as described in 'Materials and methods'. Peptides were confirmed in the HDX-MS experiment. Blue bar, the covering of each peptide. The amino-acid sequencing identification number should be –1 for each position due to the processed Met at position 1. The 10xHis Tag was included in the data analysis. (**b**) Residual plots (D_Nb725_4-bound - Nb-free) against the overlapping peptide numbers for each time point and the sum of uptake. MelB_St alone or bound with Nb725_4 in the presence of Na+ were used to carry out the HDX reactions as described in 'Materials and methods'. Black, cyan, and blue bars, the deuterium uptake at 30, 300, and 3000 s, respectively; gray curve, the sum of uptake from all three time points. Deprotection, ΔD (D_Nb725_4-bound – Nb-free) > 0; protection, ΔD < 0. Each sample was analyzed in triplicates. Cylinders indicate the helices and the transmembrane helices are labeled in Roman numerals. The length of the cylinder does not reflect the length of corresponding helices but is estimated for locations of the deuterium-labeled overlapping peptides. Noteworthy, the uncovered regions were not included. ML (cytoplasmic middle loop) and C-terminal tail including the His tag are colored in yellow. Dashed lines, the threshold.

The online version of this article includes the following source data for figure 5:

**Source data 1.** Output results of HDExaminer software analysis for *Figures 5 and 6*.

**Table 4.** HDX reaction and labeling details.

| Samples measured | WT MelB$_{St}$ | WT MelB$_{St}$ complexed with Nb725m |
|---|---|---|
| HX reaction buffer | 25 mM Tris-HCl, pD 7.5, 150 mM NaCl, 10% Glycerol, and 0.01% DDM | |
| Reaction temperature (°C) | 20 | |
| HX time course (s) | 0, 30, 300, 3000 | |
| Number of peptides | 153 | |
| Sequence coverage by labeling | 86% | |
| Mean peptide length | 9.3 | |
| Average redundancy | 2.9 | |
| Replicates (technical) | 3 | |
| \|ΔD\| (Da) | 0.3071 | |
| Back exchange rate | Not appliable | |

or poorly resolved. The deuterium uptakes in the Nb725_4-bound MelB$_{St}$ complex likely result from structural flexibility.

Group III peptides show higher-level HDX exchange rates of greater than 60% for both bound and free states even at 30 s (*Figure 6*, drawing in yellow). Nb725_4 only affords short-time, weak protections. They are located in the C-terminal tail unstructured region, and most of them are structurally unresolved due to severe disorders. Group VI peptide only contains the short N-terminal tail showing a medium-level uptake (40–50%) across all labeling times, and Nb725_4 affords a slight deprotection due to allosteric changes upon the Nb725_4 binding (*Figure 6*, cartoon and curve in red). The results show that groups III and IV peptides covering both tails are solvent-accessible dynamic regions, which is consistent with the notion that they play little role in transport-critical conformational transitions.

The 30 s deuterium uptake in the absence of Nb725_4 for the first two groups was further analyzed. Among the five group I and four group II members with faster exchanges as judged by the shortest reaction time of 30 s (*Figure 6—figure supplement 3*), except for the two periplasmic loop$_{7-8}$ (peptide 261–271) and loop$_{9-10}$ (peptide 317–326), others are clustered on the cytoplasmic side. In the outward-facing structure (*Figure 7a*), the spotted peptides pack against one another: helix-V (peptide 137–150) with helix-VIII (peptide 284–289), and the middle-loop helix (peptide 220–233) with helix-X (peptide 364–368). All surround the conformation-dependent charged network. Notably, this short peptide 137–150 links the sugar-binding residue Arg149 to the charged network by residues Lys138, Arg141, and Glu142 (*Figure 4b*). In the inward-facing structure (*Figure 7b*), these packed helices were moved against one another. Thus, these higher dynamic regions may play important roles in responding to the ligand binding and initiating the global conformational transition.

## Discussion

Formation and stabilization of a high-energy inward-facing conformation of MelB$_{St}$ in a form suitable for structure determination by cryoEM-SPA was a technically challenging sample preparation problem, which we solved by the formation of a complex of MelB$_{St}$ with a conformation-selective binder and fiducial NabFab. The resulting structure, a high-energy inward-facing, low sugar-affinity state of MelB$_{St}$ with a bound Na$^+$, represents the intracellular sugar-releasing/rebinding state in the stepped-binding model for the mechanism of coupled transport (*Figure 8*; *Guan, 2018*; *Guan and Hariharan, 2021*). Our previous results on the cooperativity of binding (*Hariharan and Guan, 2021*) supported the contention that Na$^+$ binding increases the melibiose affinity to allow melibiose uptake at lower available concentrations, a critical evolved mechanism for cation-coupled transporters. The new Na$^+$-bound sugar-releasing inward-facing cryoEM structure provides a structural basis to support the transport model based on a sequential process of the sugar-binding after Na$^+$ binding first and sugar release prior to Na$^+$ release (*Figure 8*).

The alternating-access movement has been widely used to describe the conformational change of transporters (*Abramson et al., 2003*; *Huang et al., 2003*; *Boudker et al., 2007*; *Yan, 2015*; *Drew*

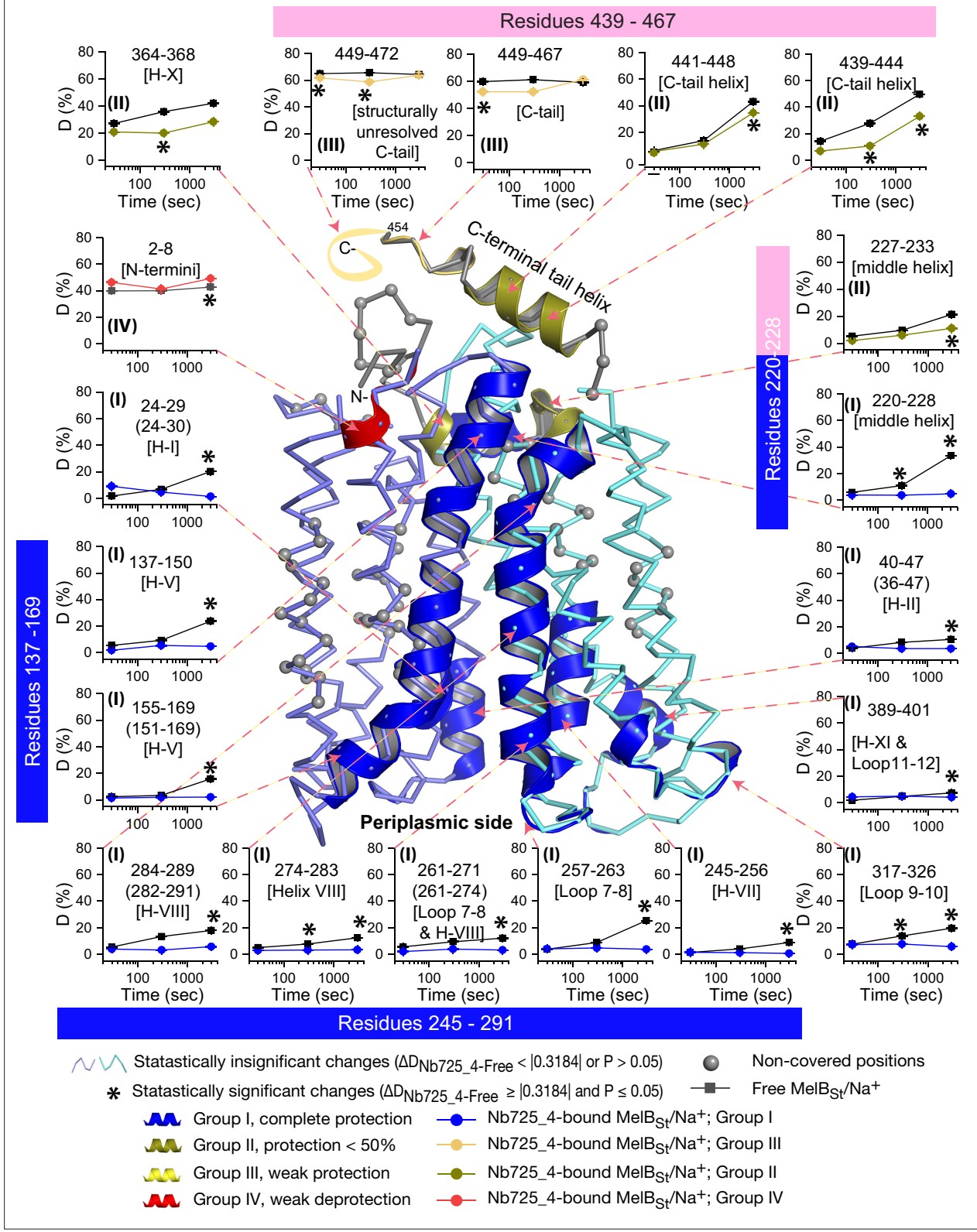

**Figure 6.** Peptide mapping of hydrogen/deuterium exchange (HDX) results. HDX results are presented in *Figure 5*. Any peptide of ΔD with p≤0.05 and ΔD ≥ |0.3184| were treated as significant. The peptides with statistically significant differences at the 3000 s time point were mapped on the outward-facing structure (PDB ID 7L17). Inset, the deuterium uptake time course of representative peptides was plotted as a percentage of deuterium uptake relative to the theoretic maximum number (D%). The peptides, either as a single peptide or a group of overlapping peptides (in the parentheses), are

*Figure 6 continued on next page*

*Figure 6 continued*

labeled and indicated in the structure by the pink arrow in dashed lines. In the brackets, the corresponding secondary structure or loops. Error bar, SEM; the number of tests, 3. Other symbols are presented within the figure.

The online version of this article includes the following figure supplement(s) for figure 6:

**Figure supplement 1.** Uptake time course for positions 2–263.

**Figure supplement 2.** Uptake time course for positions 261–475.

**Figure supplement 3.** Hydrogen/deuterium exchange mass spectrometry (HDX-MS).

*et al., 2021*; *Guan and Kaback, 2006*; *Khare et al., 2009*). This description focuses on how the substrates access their binding site or leave the protein. The core symport mechanism for coupled substrate translocations is the mobile barriers involving the inner and outer barriers as proposed originally by Peter Mitchell (*Mitchell, 1990*; *Figure 4a and d*). Our binding thermodynamic cycle study on the determination of heat capacity change ($\Delta C_p$) (*Hariharan and Guan, 2021*) showed positive values with the binding of one substrate, suggesting the dominance of hydrations and opening of the transporter, and negative values with the second substrate binding, suggesting dehydration and transition to the occluded intermediate state as indicated at step (3) in *Figure 8*. The data fit well with a barrier mechanism that prevents a single substrate molecule from crossing the protein to favor obligatory co-translocation. When this inner barrier breaks as a result of the conformational switch induced by the binding of two substrates, the sugar-binding specific pocket which is part of this inner barrier also breaks (*Figure 4e*), creating the intracellular sugar low-affinity state and the sugar-releasing path. Collectively, based on structural and thermodynamical analyses, all show that cooperative binding is integral to barrier switching in the core symport mechanism.

For substrate translocation, historically, a two-binding-site walking model was proposed: a high-affinity site at the outer surface and a low-affinity site at the inner surface. In this functional and structural paradigm, a substrate molecule moves across the protein from its high-affinity binding site to a low-affinity site for subsequent release. This canonical model has been challenged in LacY as the same binding affinity was measured at both the inner and outer surfaces (*Guan and Kaback, 2004*), and a single substrate-binding site was also observed in its 3D structures (*Abramson et al., 2003*; *Huang et al., 2003*; *Guan and Kaback, 2006*). Therefore, a single-binding site in the middle of the protein has been the current canonical model. Actually, one of the major challenges in measuring the substrate binding in membrane transporters is the intrinsic dynamic feature of the transport proteins, which makes it difficult to attribute the outcome to a specific state. Therefore, the substrate binding affinity measured in a bulk sample is an average of binding affinities from all of the individual molecules in a distribution. It is also challenging to establish a correlation between measured binding affinities and experimentally determined structures. We approached this problem by using the conformation-specific binding protein Nb725_4 to decouple the binding and conformational dynamics shifting conformers to a single state or a narrowed cluster, that is, through experimental modulation of the distribution. The results of the structural determination and binding analysis in the combination of the conformation-selective Nb permitted the extension of the current single substrate-binding site (and single affinity) model to a more refined model of one site possessing conformation-dependent higher-affinity and lower-affinity states. Identification of this critical lower-affinity inward-open state of the sugar-binding site is significant for understanding the molecular basis of higher intracellular substrate accumulation.

The shared cation site for $Na^+$, $Li^+$, or $H^+$, which is hosted only by the N-terminal two helices (II and IV), is in close proximity but not part of the mobile barrier at either outward- and inward-facing state. The inward-facing state retains the $Na^+$ binding affinity (*Figure 1—figure supplement 3d*, *Table 2*) and the bound $Na^+$ ion was observed (*Figure 3a and b*). Consistently, the simulated $Na^+$-binding site at both outward- and inward-facing conformations is virtually identical (*Figure 3—figure supplement 1*), which is in stark contrast to the conformation-dependent sugar binding in MelB$_{St}$. No conformational dependency of $Na^+$ binding might suggest the mechanism for the reversible reaction. It is likely that this $Na^+$-binding site co-evolved with the *E. coli* intracellular $Na^+$ homeostasis maintained to 3–5 mM (*Hunte et al., 2005*) to achieve optimal activity.

To interrogate the structural dynamics of MelB$_{St}$ and conformational switch mechanisms, an HDX-MS comparability study in the absence or presence of the conformational Nb725_4 was performed.

HDX-MS is a powerful approach for characterizing protein conformational flexibility and dynamics (*Masson et al., 2019*; *Zheng et al., 2019*) and has recently been applied to membrane transport proteins (*Zheng et al., 2017*; *Jia et al., 2020*; *Zmyslowski et al., 2022*). Under experimental conditions, two major parameters (solvent accessibility and hydrogen protonation) contribute to the H/D exchange rate. The residual plot clearly indicates that the major effects of Nb725_4 binding are the reduction of the conformational dynamics of MelB$_{St}$ (*Figure 5b*). Stronger protections are observed at both transmembrane helix bundles, especially both ends of the cavity-lining helices (*Figure 6*, cartoon in blue), which marks the most conformationally dynamic regions that correlate to the conformational transition in MelB$_{St}$. Notably, the Nb-free Na$^+$-bound WT MelB$_{St}$ exists in both outward- and inward-facing conformations with a bias toward the outward-facing state (*Ethayathulla et al., 2014*; *Guan and Hariharan, 2021*), and the MelB$_{St}$ in the Nb725_4 complex exists only as an inward-facing conformational ensemble. Therefore, the differences observed by HDX-MS between the two samples in the absence and presence of Nb725_4 likely disclosed, to some extent, the MelB$_{St}$ global conformational transitions, in addition to the atomic motions.

While the resolution of HDX is at a peptide level, the data showed marked effects on two well-conserved cytoplasmic helices of MFS transporters (on the middle loop and the C-terminal tail). Those peripheral amphiphilic helices are likely to play important roles in the stability of the inner barrier. Consistently, a recent atomic force microscopy study has shown that the middle loop becomes loosely packed in the presence of both sugar and cation (*Blaimschein et al., 2023*).

As described, the peptide 137–150 of helix V contains a sugar-binding residue Arg149 and three conformation-critical residues Lys138, Arg141, and Glu142 in the inner barrier-specific charged network (*Figure 4*). Extensive site-directed mutagenesis and second-site suppression showed that this charged network play important role in the conformational changes and the mutations at all three Arg at positions 141, 295, and 363 (*Figure 4b*) can be substantially rescued by a single mutation D35E (helix I) located at the other side of the membrane (*Amin et al., 2014*). Arg141 and Glu142 of

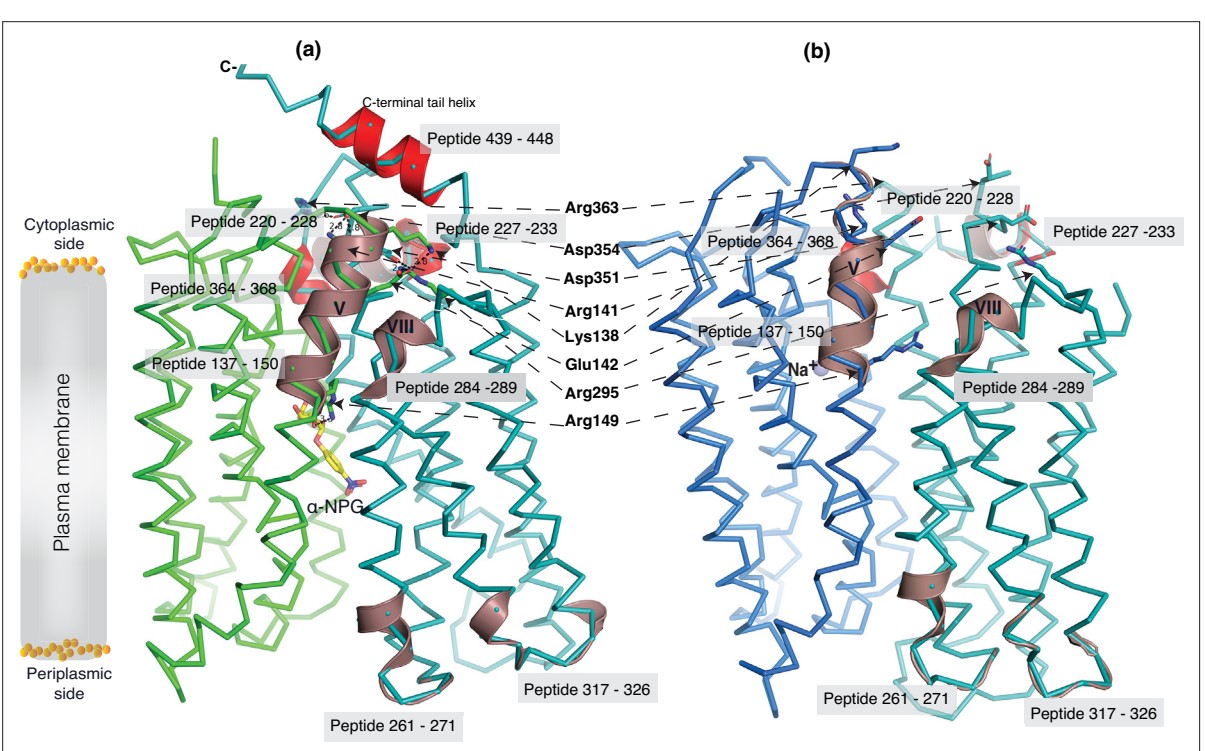

**Figure 7.** Dynamic regions of MelB$_{St}$. The peptides that exhibited faster hydrogen/deuterium exchange (HDX) rates (>5% at 30 s) were mapped on both the α-NPG-bound outward-facing structure (7L17) in panel (**a**) and the inward-facing cryoEM Nb725_4-bound structure in the panel (**b**), respectively. All peptides are labeled and highlighted in a transparent gray box. The charged residues forming the inner barrier-specific salt-bridge network are highlighted in sticks and indicated by black arrows in dashed lines. In the Nb725_4-bound structure (**b**), the C-terminal tail was disordered. The cartoon colored in dirty violet, the group I peptides with faster HDX rate; the cartoon colored in red, group II peptides. The α-NPG is highlighted in yellow and Na$^+$ is shown in blue sphere.

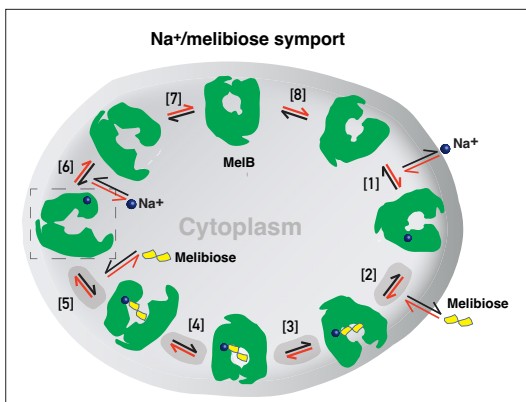

**Figure 8.** Stepped-binding model for the Na$^+$/melibiose symport catalyzed by MelB. Eight states are postulated including transient intermediates. In this reversal reaction, the cation binds prior to the sugar and releases after sugar release. Melibiose active transport or inflex begins at step [1] and proceeds via the red arrows clockwise around the circle, with one melibiose and one cation inwardly across the membrane per cycle. Melibiose efflux begins at step [6] and proceeds via the black arrows anticlockwise around the circle, with one melibiose and one cation outwardly across the membrane per circle. Melibiose exchange begins at step [6], and only takes four steps involving steps [2–5] as highlighted in gray color. The low-sugar affinity inward-facing Na$^+$-bound cryoEM structure represents the state after the release of sugar, as indicated by a dashed box.

*E. coli* MelB have been shown to participate in the conformational transition (*Meyer-Lipp et al., 2006*). Pro148 has been identified by second-site mutation to reduce the transport $V_{max}$ of a $V_{max}$ elevated mutant by modulating the rate of conformational change (*Jakkula and Guan, 2012*). Thus, the helix-V dynamic region (peptide 137–150) could link the sugar-binding pocket with the inner barrier-specific salt-bridge network. Together with the interactions with the dynamic regions (residues 284–289) of helix VIII, the substrate binding occupancy could be propagated, via an allosteric mechanism, from the sugar-binding pocket to the cytoplasmic peripheral helices as a pre-transition for the opening of the transporter to the cytoplasm (*Blaimschein et al., 2023*).

All of the conformation-selective Nbs (Nb725, Nb733, and the hybrid Nb725_4) are expected to bind MelB$_{St}$ similarly and trap it at the high-energy state, which is consistent with the observation of very low-frequency occurrence during Nbs selection (*Katsube et al., 2023*). The effects of the Nbs binding mimic the regulatory effect imposed by EIIA$^{Glc}$ binding (*Hariharan and Guan, 2014*; *Hariharan et al., 2015*), that is, reduction of sugar-binding affinity with no change in the Na$^+$ binding. Interestingly, either Nbs or EIIA$^{Glc}$ can bind to MelB$_{St}$ in the absence or presence of the other (*Table 2*, *Figure 1—figure supplements 2 and 3*); the complex composed of all three proteins can be isolated (*Figure 2—figure supplement 8*; *Katsube et al., 2023*). The extensive comparisons strongly support the notion that the Nbs and EIIA$^{Glc}$ can concurrently recognize two spatially distinct epitopes of MelB$_{St}$, and the inward-facing conformation trapped by the Nb mimics the conformation of MelB$_{St}$ under the EIIA$^{Glc}$ regulation. Available data showed that three positions (Asp438, Arg441, and Ile445) on one face of the C-terminal tail helix of MelB$_{St}$ might be involved in EIIA$^{Glc}$ binding (*Kuroda et al., 1992*), and this region is conserved with the later known EIIA$^{Glc}$ binding site in MalK (*Chen et al., 2013*). Thus, the cytoplasmic surface of the C-terminal domain could potentially support the EIIA$^{Glc}$ binding at the inward-facing state; unfortunately, the C-terminal tail helix is disordered in this Nb-bound inward-facing structure.

In this study, we demonstrated that the mobile barrier mechanism plays a central role in substrate translocation, the barrier dynamics are modulated by the cooperative binding of substrates, and the inner/outer barrier shifting directly regulates the sugar-binding affinity, with no effect on the Na$^+$ binding. Furthermore, we proposed that the functional inhibition of this MFS transporter by the central regulatory protein EIIA$^{Glc}$ is achieved by trapping MelB$_{St}$ in a low-sugar affinity inward-facing state. Overall, our studies provide substantial information to explain the obligatorily coupled transport of Na$^+$ and sugar substrate in structural and dynamic terms collectively.

## Materials and methods

### Key resources table

| Reagent type (species) or resource | Designation | Source or reference | Identifiers | Additional information |
|---|---|---|---|---|
| Strain, strain background *melB* (*Salmonella typhimurium*) | *S. typhimurium* strain LT2/ SGSC1412/ATCC | PMID:1495487 | STM4299 | Used for *melB* cloning |
| Strain, strain background (*Escherichia coli*) | DW2 | PMID:3047112 | *melA⁺ ΔmelB ΔlacZY* | MelB expression and transport analysis |
| Strain, strain background (*E. coli*) | XL1 Blue | Agilent Technologies | *recA1 endA1 gyrA96 thi-1 hsdR17 supE44 relA1 lac [F' proAB lacIqZΔM15 Tn10 (Tetr)]* | Plasmid amplification |
| Strain, strain background (*E. coli*) | ArcticExpress (DE3) | Agilent Technologies | F⁻ *ompT hsdS*(rB⁻ mB⁻) *dcm⁺* Tetr gal $\lambda$ (DE3) *endA* Hte [cpn10 cpn60 Gentr] | Protein expression |
| Strain, strain background (*E. coli*) | DH5α *cyaA⁻* | PMID:37380079 | *ΔcyaA* | Two-hybrid assay |
| Strain, strain background (*E. coli*) | BL21(DE3) T7 express | New England Biolabs | *fhuA2 lacZ::T7 gene1 [lon] ompT gal sulA11 R(mcr-73::miniTn10--Tetˢ)2 [dcm] R(zgb-210::Tn10--Tetˢ) endA1 Δ(mcrC-mrr)114::IS10* | Protein expression |
| Strain, strain background (*E. coli*) | BL21(DE3) C43 | PMID:8757792 | F⁻ *ompT hsdSB (rB- mB-) gal dcm (DE3)* | Protein expression |
| Strain, strain background (*E. coli*) | BL21(DE3) pRIL | Agilent Technologies | F⁻ ompT hsdS(rB⁻ mB⁻) dcm⁺ Tetʳ gal endA Hte [argU ileY leuW], Camʳ | Protein expression |
| Recombinant DNA reagent (plasmid) | pCS19 | PMID:10319814 | pQE60 derivative inserted with gene *lacIq*; ampʳ | Cloning/expressing vector |
| Recombinant DNA reagent (plasmid) | pCS19/FX | PMID:25627011 | Expression vector derived from pCS19 with two SapI sites and *ccdB* gene for FX cloning; ampʳ | |
| Recombinant DNA reagent (plasmid) | pACYC | PMID:2190220 | Expression vector; no *ccdB* gene: camʳ | |
| Recombinant DNA reagent (plasmid) | pACYC/FX | PMID:25627011 | Expression vector derived from pACYC; with *ccdB* gene, camʳ | |
| Recombinant DNA reagent (plasmid) | pACYC/MelB_St | PMID:25627011 | Expression plasmid for MelB_St derived from pACYC/FX; no *ccdB* gene, camʳ | |
| Recombinant DNA reagent (plasmid) | pCS19/X:T18/FX | PMID:37380079 | FX cloning vector; two SapI sites and *ccdB* gene for FX cloning; ampʳ | Two-hybrid assay vector; expressing a target protein 'X' with a C-terminal T18 fusion |
| Recombinant DNA reagent (plasmid) | pCS19/T18 | PMID:37380079 | Expression plasmid from pCS19/X:T18/FX for expressing T18 fragment only; no *ccdB* gene. | Two-hybrid assay plasmid; control vector |
| Recombinant DNA reagent (plasmid) | pACYC/T25 | PMID:37380079 | Expression plasmid from pACYC/T25:X/FX for expressing T25 fragment; no *ccdB* gene, camʳ | Two-hybrid assay plasmid; control vector |
| Recombinant DNA reagent (plasmid) | pACYC/T25:MelB_St | PMID:37380079 | Expression plasmid derived from pACYC/T25:X/FX; no *ccdB* gene, camʳ | Two-hybrid assay plasmid; expressing T25:MelB_St hybrid |

| Reagent type (species) or resource | Designation | Source or reference | Identifiers | Additional information |
|---|---|---|---|---|
| Recombinant DNA reagent (plasmid) | pCS19/Nb725:T18 | PMID:37380079 | Expression plasmid hybrid derived from pCS19/X:T18/FX; no *ccdB* gene, amp$^r$ | Two-hybrid assay plasmid; expressing Nb725:T18 hybrid |
| Recombinant DNA reagent (plasmid) | pCS19/Nb725_4:T18 | This study | Expression plasmid derived from pCS19/X:T18/FX; no *ccdB* gene, amp$^r$ | Two-hybrid assay plasmid; expressing Nb725_4:T18 hybrid |
| Recombinant DNA reagent (plasmid) | pK95ΔAH/MelB$_{St}$/CHis$_{10}$ | PMC3057838 | Constitutive expression plasmid | MelB$_{St}$ protein expression |
| Recombinant DNA reagent (plasmid) | pCS19/Nb725 | PMID:37380079 | Expression plasmid for Nb725 derived from pCS19/FX; no *ccdB* gene, amp$^r$ | Nb725 protein expression |
| Recombinant DNA reagent (plasmid) | pCS19/Nb725_4 | This study | Expression plasmid for Nb725_4 derived from pCS19/FX; no *ccdB* gene, amp$^r$ | Nb725_4 protein expression |
| Recombinant DNA reagent (plasmid) | pET26b(+) | Novagen (EMD Millipore) | Periplasmic expression plasmid with a N-terminal pelB signal sequence; Kan$^r$ | For expression Nb725_4 and anti-Fab Nb |
| Recombinant DNA reagent (plasmid) | pET26/Nb725_4 | This study | Periplasmic expression plasmid derived from pET26b(+); Kan$^r$ | Nb725_4 protein production |
| Recombinant DNA reagent (plasmid) | pET26/Anti-Fab Nb | This study | Anti-Fab Nb periplasmic expression plasmid; Kan$^r$ | Anti-Fab Nb protein expression |
| Recombinant DNA reagent (plasmid) | p7XC3H/Nb725 | PMID:37380079 | Cytoplasmic expression plasmid; Kan$^r$ | Nb725 protein production |
| Recombinant DNA reagent (plasmid) | p7XNH3/EIIA$^{Glc}$ | PMID:25296751 | Cytoplasmic expression plasmid; Kan$^r$ | EIIA$^{Glc}$ protein production |
| Recombinant DNA reagent (plasmid) | pR2.2/NabFab | PMID:34782475 | Periplasmic expression of NabFab; Amp$^r$ | NabFab protein production |
| Recombinant DNA reagent (plasmid) | pMSP1E3D1 | Addgene/20066 | Expressing SP1E3D1; Kan$^r$ | MSP1E3D1 production |
| Recombinant DNA reagent (plasmid) | pRK792 | Addgene/8830 | Expressing TEV protease; Amp$^r$ | TEV protease production |
| Sequence-based reagent (primers) | MelB_Nb725_4_T18 | This study | Fwd: 5'- ATATATGCTCTTCTAGTCAACGTCAATTGGTAG -3' Rev: 5'- TATATAGCTCTTCATGCGCTGCTCACGGTCAC -3' | FX cloning primers to contrast pCS19/Nb725_4:T18; addition of a C-terminal 'A' of Nb725_4 enabling the C-terminal T18 in-frame fusion |
| Sequence-based reagent (primer) | MelB_Nb725_4 | This study | Fwd: 5'-ATATATGCTCTTCTAGTATGCAACGTCAATTGGTAG-3' Rev: 5'-TATATAGCTCTTCATGCTTAGTGGTGATGATGGTGG TGGCT GCTCACGGTCAC-3' | FX cloning primers to construct pCS19:Nb725_4-CTH with a C-terminal 6x His-tag |
| Sequence-based reagent (primer) | Nb-PelB-NdeI | This study | Fwd: 5'-TTTAAGAAGGAGATATACATATG-3' | Insert NdeI restriction site for Nb725_4 and anti-Fab Nb construction |

| Reagent type (species) or resource | Designation | Source or reference | Identifiers | Additional information |
|---|---|---|---|---|
| Sequence-based reagent (primer) | Nb-STRP-XhoI | This study | Rev: 5'-TTTGTTCTAGACTCGAGTTATTTCTC-3' | Add XhoI restriction site for Nb725_4 and anti-Fab Nb construction |
| Chemical compound, drug | [³H]Melibiose | PerkinElmer | (5.32 Ci/mmol) | Transport assay S |
| Chemical compound, drug | UDM | Anatrace | Cat# D300HA | MelB$_{St}$ purification |
| Chemical compound, drug | DDM | Anatrace | Cat# D310 | MelB$_{St}$ purification |
| Chemical compound, drug | Melibiose | Acros Organics | Cat# 125375000 | Fermentation and Binding assay |
| Chemical compound, drug | α-NPG | Acros Organics | Cat# 33733500 | Binding assay |
| Chemical compound, drug | E. coli lipids | Avanti Polar Lipids, Inc | Extract Polar, 100600 | Nanodiscs |
| Software, algorithm | CryoSPARC | CryoSPARC | 2-4.01 | CryoEM data processing |
| Software, algorithm | USF ChimeraX | USF ChimeraX | 1.6 | Mask preparation |
| Software, algorithm | Phenix | Phenix | 1.20-4459 | Map sharpen and model refinement |
| Software, algorithm | Coot | Coot | 0.9 | Model building |
| Software, algorithm | Pymol | Pymol | 2.5 | Model visualization |
| Software, algorithm | Qscore | Qscore | | Q score calculation |
| Software, algorithm | NanoAnalyze | TA Instruments | 3.7.5 | Data fitting |
| Software, algorithm | HDExaminer | Trajan Scientific and Medical | 3.3 | HDX data analysis |
| Software, algorithm | BioPharma Finder | Thermo | 5.1 | HDX data analysis and peptide mapping |

## Reagents

MacConkey agar media (lactose-free) was purchased from Difco. Unlabeled melibiose and 4-nitrophe nyl-α-D-galactopyranoside (α-NPG) were purchased from Sigma-Aldrich. Maltose was purchased from (Acros Organics, Fisher Scientific). [1-³H]Melibiose (5.32 Ci/mmol) was custom synthesized by Perki-nElmer. Detergents undecyl-β-D-maltopyranoside (UDM) and n-dodecyl-β-D-maltopyranoside (DDM) were purchased from Anatrace. E. coli lipids (Extract Polar, 100600) were purchased from Avanti Polar Lipids, Inc,. Oligodeoxynucleotides were synthesized by Integrated DNA Technologies. LC-MS grade water, LC-MS 0.1% formic, LC-MS grade acetonitrile with 0.1% formic acid in water were purchased from Fisher Scientific (Hampton, NH). Guanidine hydrochloride, citric acid, and zirconium (IV) oxide 5 µm powder were purchased from Sigma-Aldrich (St. Louis, MO). Deuterium oxide (99⁺% D) was purchased from Cambridge Isotope Laboratories (Tewksbury, MA). All other materials were reagent grade and obtained from commercial sources.

## Strains, plasmids, and primers

The genotype and source of *E. coli* strains used in this study are described in Key Resources Table unless otherwise described specifically. The commercial *E. coli Stellar or XL1 Blue* were used for plasmid construction. The *E. coli* DB 3.1 strain was used to construct the *ccdB*-containing universal FX Cloning vectors (*Geertsma and Dutzler, 2011*). *E. coli* DW2 (*melB⁻lacY⁻*) (*Botfield and Wilson, 1988*) was used for functional studies, and DH5α *cyaA⁻* strain was used for in vivo protein–protein interaction assay (*Katsube et al., 2023*). *E. coli* BL21 (DE3) strain and *E. coli* ArcticExpress (DE3) strain were used for overexpression Nbs; *E. coli* BL21 (DE3) C43 strain (*Miroux and Walker, 1996*) was used for overexpression NabFab; *E. coli* BL21(DE3) T7 express strain was used to express EIIA^Glc (*Hariharan and Guan, 2014*). The plasmids used or created and primers in this study are also listed in the Key Resources Table.

## CDR grafting

CDR grafting technique (*Riechmann et al., 1988*) to transfer the binding specificity of MelB$_{St}$ Nbs to the TC-Nb4, which can be recognized by the NabFab, was performed (*Bloch et al., 2021*; *Figure 1— figure supplement 1*). The DNA fragment, which carried the hybrid Nb725_4 along with N-terminal pelB signal peptide (MKYLLPTAAAGLLLLAAQPAMA) and C-terminal HRV-3C protease site (LEVLFQGP) and 8xHis-tag, was synthesized, digested using restriction enzymes NdeI and XhaI, and ligated with the corresponding sites on expression plasmid pET26b⁺, resulting in the pET26/Nb725_4.

## MelB$_{St}$ protein expression and purification

The MelB$_{St}$ expression plasmid pK95ΔAH/MelB$_{St}$/CHis$_{10}$ (*Guan et al., 2011*) was constructed by cloning *melB* from *Salmonella* typhimurium LT2 strain (*Mizushima et al., 1992*) for constitutive expression in *E. coli* (*Pourcher et al., 1995*). A 10 L fermenter was used for the constitutive expression of MelB$_{St}$ in DW2 cells in Luria-Bertani (LB) broth supplemented with 50 mM KPi, pH 7.0, 45 mM (NH$_4$) SO$_4$, 0.5% glycerol, and 100 mg/L ampicillin (*Ethayathulla et al., 2014*). The cultures were grown at 30°C until $A_{280}$ ~5. The procedures for membrane preparation, extraction with 1.5% UDM or 2% DDM, and cobalt-affinity purifications were performed as previously described (*Ethayathulla et al., 2014*; *Hariharan and Guan, 2017*; *Katsube et al., 2023*). The eluted MelB$_{St}$ proteins were dialyzed overnight against a buffer of 20 mM Tris–HCl, pH 7.5, 100 mM NaCl, 10% glycerol, and 0.35% UDM or 0.01% DDM, and concentrated with Vivaspin 50 kDa MWCO membrane to approximately 50 mg/mL, stored at –80°C after flash-frozen with liquid nitrogen.

## Overexpression and purification of membrane scaffold protein 1E3D1 (MSP1E3D1)

An expression plasmid pMSP1E3D1 (Addgene; plasmid 20066) was used for the overexpression of the MSP1D1E3 with a 7-His-tag and a TEV protease site in the *E. coli* BL21 (DE3) (*Denisov et al., 2007*; *Denisov and Sligar, 2017*). The culture was grown in LB media containing 0.5% glucose at 37°C, induced by adding 1 mM IPTG at $A_{600}$ of about 0.6, and incubated for another 2.5 hr. The MSP proteins from the cell lysate were purified with INDIGO Ni-Agarose (Cube Biotech) using a protocol as described (*Hariharan and Guan, 2021*; *Zoghbi et al., 2016*). The eluted proteins were dialyzed against 20 mM Tris–HCl, pH 7.5, and 100 mM NaCl and concentrated to ~8 mg/mL. The His-tag of MSP1E3D1 was removed by a His-tagged TEV protease (at 1:20 mole/mole ratio of TEV:MSP1E3D1) in the same buffer. The His-tagged TEV protease, unprocessed His-tagged MSP1E3D1, and free His tag fragments were trapped within the Ni-Agarose column. The processed MSP1E3D1 proteins without a His tag in the flow-through were concentrated to about 6–8 mg/ml, and stored at –80°C after flash-frozen with liquid nitrogen.

## TEV protease expression and purification

The tobacco etch virus (TEV) encoded by the plasmid pRK792 (*Kapust et al., 2001*) was overexpressed in a tRNA-supplemented *BL21*-CodonPlus (DE3)-RIPL strain. The cells were grown in LB broth containing 100 μg/mL ampicillin and 30 μg/mL chloramphenicol at 30°C until $A_{600}$ of 0.8~1.0. At this point, the temperature was decreased to 18°C, and protein expression was induced by 0.5 mM IPTG overnight. The cells in 50 mM KPi, pH 8.0, 500 mM NaCl, and 10% glycerol were lysed by a microfluidizer and the supernatants after removing the cell debris were subjected to affinity chromatography

using Talon resin. The eluted proteins were concentrated to ~3 mg/mL and quickly dialyzed against 20 mM Tris–HCl, pH 7.5, 200 mM NaCl, and 10% glycerol for 3–4 hr, and stored at –80°C after flash-frozen with liquid nitrogen. A typical yield can reach >30 mg/L.

## MelB$_{St}$ reconstitution into lipids nanodiscs

The MelB$_{St}$ reconstitution was performed as described previously (*Hariharan and Guan, 2021*). Briefly, 5.6 mg of *E. coli* polar lipids extract at a concentration of 40 mg/mL solubilized in 7.5% DDM was added to 1 mL of 1 mg of MelB$_{St}$ in 20 mM Tris–HCl, pH 7.5, 100 mM NaCl, 10% glycerol, and 0.35% UDM (~1:350 mole/mole), and incubated for 10 min on ice. MSP1E3D1 proteins were then added into the MelB$_{St}$/lipid mixture at a 5:1 mole ratio (MSP1E3D1:MelB$_{St}$), and incubated at room temperature with mild stirring for 30 min before being shifted to 4°C with mild stirring. The detergents were removed and subsequent MelB$_{St}$ lipids nanodiscs were formed by stepwise additions of Bio-beads SM-2 resin (500 mg and then an additional 300 mg after 2 hr) and further incubated overnight. After separating the Bio-beads SM-2 resins, the MelB$_{St}$ lipids nanodiscs were isolated by absorbing onto Ni-NTA beads and the MelB$_{St}$-free empty nanodiscs were in the void due to His-tag being removed. The MelB$_{St}$ lipids nanodiscs were eluted by adding 250 mM imidazole and the elutes were dialyzed against 20 mM Tris–HCl, pH 7.5, and 150 mM NaCl, and concentrated to ~3–4 mg/mL. After ultra-centrifugation at ~352,000 × $g$ (90,000 rpm using a TLA-120 rotor) in a Beckman Optima MAX Ultra-centrifuge for 30 min at 4°C, the supernatant containing MelB$_{St}$ lipids nanodiscs were stored at –80°C after flash-frozen with liquid nitrogen.

## Nbs expression and purification

*E. coli* BL21(DE3) Arctic express cells were used to produce Nbs. The Nb725_4 and anti-Fab Nb were expressed by the expression plasmid pET26/Nb725_4 and pET26/anti-Fab Nb, respectively, containing the *pelB* leader sequence. Nb725 was expressed by the plasmid p7xC3H/Nb725 as described (*Katsube et al., 2023*). Cells were grown in Terrific Broth media supplemented with 0.4% (vol/vol) glycerol at 37°C. Protein expression was induced by adding 1 mM IPTG at the mid-log phase after decreasing the temperature to 25°C, and the cultures continued growth for 18–20 hr. The peri-plasmic extraction of Nbs using sucrose-mediated osmolysis was as described (*Kariuki and Magez, 2021*), and Nb purification was performed by cobalt-affinity chromatography, which yielded ~5 mg/L with >95% purity. The cytoplasmic expression for the Nb725 also provided a similar yield and purity, and all were stable. The purified Nbs were concentrated to ~5–10 mg/mL, dialyzed against 20 mM Tris–HCl, pH 7.5, and 150 mM NaCl, and frozen in liquid nitrogen and stored at –80°C.

## NabFab expression and purification

The NabFab expressing plasmid pRH2.2/NabFab and the expression protocol were kindly gifted from Dr. Kasper P. Locher (*Bloch et al., 2021*; *Hornsby et al., 2015*). Briefly, the NabFab was expressed in *E. coli* C43 (DE3) cells in an autoinduction medium at 25°C for ~18 hr. The cells were resuspended in 20 mM Tris–HCl, pH 7.5, 500 mM NaCl, and 1 mM EDTA, and lysed by a microfluidizer. The lysates were subjected to heat treatment at 65°C for 30 min, followed by centrifugation at 33,175 × $g$ (15,000 rpm on the rotor JA18) for 30 min at 4°C to remove insoluble materials. The supernatants were loaded onto the 5 mL Cytivia Protein L column for affinity purification of NabFab proteins. The eluted NabFab in 0.1 M acetic acid, pH 2.0, was neutralized with 200 mM Tris–HCl, pH 8.5, dialyzed against 20 mM Tris–HCl, pH 7.5, 150 mM NaCl, and concentrated up to 5 mg/mg prior to being frozen in liquid nitrogen and stored at –80°C. The yield for NabFab protein is about 2–3 mg/L.

## EIIA$^{Glc}$ expression and purification

The overexpression of unphosphorylated EIIA$^{Glc}$ was performed in the *E. coli* T7 Express strain (New England Biolabs) as described (*Hariharan and Guan, 2014*).

## A bacterial two-hybrid assay

A bacterial two-hybrid assay based on reconstituting adenylate cyclase activities from the T18/T25 fragments of *Bordetella pertussis* adenylate cyclase toxin as described (*Battesti and Bouveret, 2012*; *Ladant and Ullmann, 1999*) was used to examine in vivo interaction of Nb and MelB$_{St}$ in *E. coli* DH5α *cyaA* strain (*Katsube et al., 2023*). The competent cells were transformed by two compatible plasmids

(pACYC [*Bibi and Kaback, 1990*] and pCS19 [*Spiess et al., 1999*]) with different replication origins (e.g., pACYC/T25:MelB$_{St}$ vs. pCS19/Nb725:T18 [*Katsube et al., 2023*] or pCS19/Nb725_4:T18), and plated onto the lactose-free MacConkey agar plate containing 30 mM maltose as the sole carbohydrate source, 100 mg/L ampicillin, 25 mg/L chloramphenicol, and 0.2 mM IPTG, and the plates were incubated in 30°C for 7 d before photography. Red colonies grown on the MacConkey agar indicated positive fermentation due to the cAMP production derived from the interaction of two hybrids. Yellow colonies denoted no fermentation and no cAMP production.

## Sugar fermentation assay

*E. coli* DW2 cells (Δ*melB*Δ*lacYZ*) were transformed with two compatible plasmids with different replication origins (pACYC/MelB$_{St}$ and pCS19/Nbs) (*Tikhonova et al., 2015*) and plated onto the lactose-free MacConkey agar plate containing 30 mM maltose or melibiose as the sole carbohydrate source, 100 mg/L ampicillin, 25 mg/L chloramphenicol, and 0.2 mM IPTG. The plates were incubated at 37°C for 18 hr before photography. Magenta colonies, a normal sugar fermentation; yellow colonies, no sugar fermentation. The cells carrying two empty plasmids or with MelB$_{St}$ but no Nb were used as the control. Maltose fermentation was another control for the specificity of Nb inhibition.

## [³H]Melibiose transport assay and western blot

*E. coli* DW2 cells (*melA$^+$, melB$^-$, lacZ$^-$Y$^-$*) (*Pourcher et al., 1995*) were transformed with the two compatible plasmids pACYC/MelB$_{St}$ and pCS19/Nb725 or pCS19/Nb725_4, respectively. The cells transformed with two empty vectors (pACYC and pCS19) or pACYC/MelB$_{St}$ with pCS19 without an Nb were used as the negative or positive control, respectively. The cells were grown in LB broth with 100 mg/L ampicillin and 25 mg/L chloramphenicol in a 37°C shaker overnight, and inoculated by 5% to fresh LB broth with 0.5% glycerol, 100 mg/L ampicillin, 25 mg/L chloramphenicol, and 0.2 mM IPTG. The cultures were shaken at 30°C for 5 hr. The transport assay was performed at 20 mM Na$^+$ and 0.4 mM [³H]melibiose (specific activity of 10 mCi/mmol) in 100 mM KP$_i$, pH 7.5, 10 mM MgSO$_4$ at $A_{420}$ of 10 (~0.7 mg proteins/mL) as described (*Guan et al., 2011*; *Jakkula and Guan, 2012*). The transport time courses were carried out at 0, 5 s, 10 s, 30 s, 1 min, 2 min, 5 min, 10 min, and 30 min by dilution and fast filtration. The filters with trapped cells were subjected to radioactivity measurements using a Liquid Scintillation Counter. The MelB$_{St}$ membrane expression was analyzed by western blot using HisProbe-HRP Conjugate as described (*Katsube et al., 2023*). The western blot result was imaged by the ChemiDoc MP Imaging System (Bio-Rad).

## Isothermal titration calorimetry

All ITC ligand-binding assays were performed with the Nano-ITC device (TA Instruments) (*Hariharan and Guan, 2017*) and the heat releases are plotted as positive peaks. All samples were buffer-matched by simultaneously dialyzed in the same buffer or diluting the concentrated small molecule ligands to corresponding dialysis buffers as described (*Hariharan and Guan, 2017*; *Hariharan and Guan, 2014*). The titrand (MelB$_{St}$) placed in the ITC Sample Cell was titrated with the specified titrant (placed in the Syringe) by an incremental injection of 1.5–2 µL aliquots at an interval of 300 s at a constant stirring rate of 250 rpm. All samples were degassed by a TA Instruments Degassing Station (model 6326) for 15 min prior to the titration. Heat changes were collected at 25°C, and data processing was performed with the NanoAnalyze (v. 3.7.5 software) provided with the instrument. The normalized heat changes were subtracted from the heat of dilution elicited by the last few injections, where no further binding occurred, and the corrected heat changes were plotted against the mole ratio of titrant versus titrand. The values for the binding association constant ($K_a$) were obtained by fitting the data using the one-site independent-binding model included in the NanoAnalyze software (version 3.7.5), and the dissociation constant ($K_d$) = 1/$K_a$. In most cases, the binding stoichiometry ($N$) number was fixed to 1.

## Protein concentration assay

The Micro BCA Protein Assay (Pierce Biotechnology, Inc) was used to determine the protein concentration assay. The $A_{280}$ was used for concentration estimation.

## Protein complex preparation for grid vitrification

Proteins NabFab, Nb725_4, and anti-Fab Nb were mixed at a mole ratio of 1:1.1:1.15. The final complex was obtained by adding WT MelB$_{St}$ in nanodiscs based on a mole ratio of 1:1.6 of MelB$_{St}$:NabFab

complex. The sample was expected to contain 6 μM MelB$_{St}$, 12 μM MPS1E3D1, 9.5 μM NabFab complex in 20 mM Tris–HCl, pH 7.5, 150 mM NaCl (~1.5 mg/mL). The complex was also analyzed by gel-filtration chromatography using GE Superdex 200 column on a Bio-Rad NGC Chromatography System. Peak fractions were examined by SDS-15% PAGE stained by silver nitrate.

## Grids vitrification and single-particle cryoEM data acquisition

Grids preparation and data collection were carried out in S$^2$C$^2$ (Menlo Park, CA). The WT MelB$_{St}$/Nb725_4/NabFab/anti-NabFab Nb complex was thawed out from –80°C storage. Aliquots of 3 μL of samples at 0.75 mg/mL or 1.5 mg/mL protein concentration were placed on 300-mesh Cu holey carbon grids (Quantifoil, R1.2/1.3) after glow-discharge (PELCO easiGlow) at 15 mA for 30 s and blotted on two sides for 3 s at 4°C and 100% humidity before vitrified in liquid ethane using a Vitrobot Mark IV (Thermo Fisher). The grids were subsequently transferred to a Titan Krios (G3i) electron microscope (Thermo Fisher) operating at 300 kV and equipped with K3 direct electron detector (Gatan) and BioQuantum energy filter. The best grids were found from the sample at 1.5 mg/mL concentration.

A total of 14,094 and 8715 movies were automatically collected using the EPU Data Acquisition Software (Thermo Fisher) for non-tilted and 30° tilted collections, respectively. Both datasets were collected at a 0.86 Å pixel size and a dose of 50 e$^-$/Å$^{-2}$ across 40 frames (0.0535 or 0.07025 s per frame for the non-tilted and tilted data collection, respectively) at a dose rate of approximately 23.36 or 17.8 e$^-$/Å$^2$/s, respectively. A set of defocus values was applied ranging from –0.8, to 1.0, –1.2,–1.4, or –1.8 μm with an energy filter slit width of 20 eV and a 100 μm objective aperture.

## Single-particle data processing

All cryoEM single-particle data processing was performed using CryoSPARC program (v. 2–4.01) (*Figure 2—figure supplement 1*). All imported movies were aligned using defaulted Patch Motion Correction and followed with defaulted Patch contrast transfer function (CTF) Estimation. Micrographs were then curated based on CFT Estimation, astigmatism, and ice thickness, which resulted in 13,649 and 7129 micrographs for the non-tilted and 30° tilted datasets, respectively.

Initially, particle picking from the 2000 micrographs was performed using Blob Picker using 120 and 160 Å for the minimal and maximal particle diameters, respectively, in the CryoSPARC program (*Figure 2—figure supplement 2*). A total of 1,087,124 particles were selected and extracted with a box size of 256 pixels. After two runs of 2D Classification, five classes were selected as imputing templates in Template Picker. A total of resultant 9,103,229 particles from 13,649 micrographs were extracted with a box size of 384 pixels bin 92 pixels and subjected to iterative rounds of 2D Classification. The yielded 1,245,734 particles were re-extracted with a box size of 416 pixels. After further sorting using 2D Classification, the highly selected 338,883 particles, which clearly contained nanodisc-surrounding MelB$_{St}$ bound with Nb725_4/NabFab/anti-NabFab Nb, were used for Ab Initio 3D Reconstruction. One out of five Ab Initio classes contained the expected super-complex from a total of 93,693 particles, which allowed for a 3.80 Å volume to be reconstructed using Non-uniform Refinement. The 93,693 particles were then further sorted using heterogeneous refinement with input volumes created by multiclass ab initio to a subset of 41,264 particles, which allowed a 3.49 Å volume to be reconstructed using Non-uniform Refinement with a static mask prepared from one of the volumes. This map was successfully used to build the initial model of all four proteins; however, this map was anisotropic and lacked some orientations in angular space. To improve the map quality, two strategies were applied including reprocessing this dataset and performing tilted data collection. The hand of the 3.49 Å volume was changed and used to generate 50 templates.

Template Pick using the 50 templated was applied to the 13,649 micrographs, which generated 7,632,727 particles (*Figure 2—figure supplement 3*). After extraction with a box size of 384 pixels bin 92 pixels and iterative rounds of 2D Classification and Heterogeneous Refinement, 370,941 particles were selected and re-extracted with a box size of 384 pixels. This 370,941 particle set was combined with the previously identified 41,264 particle set after extraction with a box size of 384 pixels and removed duplicate, the subset of 188,566 particles exhibited slightly improved directional distribution. To further explore the first data collection, the previously rejected 2D classes were analyzed and 214,522 particles were selected based on orientation. After combined and iterative rounds of 2D Classification, Heterogeneous Refinement, and Local Refinement, a subset of the 196,254 particle set

supported a 3.39 Å volume to be reconstructed using local refinement, and the directional distribution of the map is further improved.

A tilted data collection on the same microscopy and parameter settings was performed with a 30° tilt (*Figure 2—figure supplement 3*). A total number of 7129 micrographs were obtained. The 50 templates were used for particle pick, which yielded 2,887,147 particles after curation. The particles were extracted using a box size of 384 bin 96 pixels. After three-run 2D Classification followed by three-run Heterogeneous Refinement using the 3.39 Å volumes and the other two ab initio volumes, a subset of 544,531 particles were re-extracted using a box size of 384 pixels. After further iterative rounds of 2D Classification, Heterogeneous Refinement, and Local Refinement, 182,101 particles were obtained for the tilted collection, which could support a 4.01 Å volume using Local Refinement and directional distribution focused on the missing angles in the first dataset.

A total number of 378,355 polished particles from both non-tilted and 30° tilted data acquisition were used to reconstruct a 3.40 Å volume using Local Refinement, and the 30° tilted data filled the missing particle orientations but the resolution was not improved. The particles were subjected to 3D Classifications using a target resolution of 3.3 Å and 10 classes and the highest resolution classes were collected. The final collected 296,925 particle set supported a 3.37 Å volume to be reconstructed using Local Refinement and a static mask covering MelB$_{St}$, Nb725_4, and NabFab. The reporting GSFSC resolution of 3.29 Å was calculated by Validation (FSC) using the two half maps, and the particle distribution was calculated by ThereDFSC (*Figure 2—figure supplement 4*).

The initial mask for Non-uniformed Refinement was created in UCSF ChimeraX using the volume generated from Ab Initio 3D Reconstruction which covered the full complex. The refinement masks for final-stage Local Refinement only covered MelB$_{St}$, Nb725_4, and NabFab and filtered at a low pass to 10 Å and soft padding width of 10. The 3D Classification used the full-covered mask as the solvent mask and the refinement masks in the Local Refinement as the focus mask.

Local Refinement set the initial low pass resolution of 8 Å, use of pose/shift Gaussian prior during alignment, rotation search extent of 9°, and shift search extent of 6 Å. The final 3.29 Å volume was sharpened in Phenex auto_sharpen program (*Terwilliger et al., 2018*) default setting for model refinement.

## Model building and structure refinement

The initial model building was carried out based on the 3.45 Å reconstruction map obtained from the first dataset. The coordinates of MelB$_{St}$ D59C (PDB ID 7L17), NabFab, and anti-NabFab Nb (PDB ID 7PHP), and a sequence-based 3-D model generated by AlphaFold 2 were fitted into the map in UCSF ChimeraX (*Goddard et al., 2018*). The N-terminal helices of MelB$_{St}$ fit well and the C-terminal helices were manually fitted into the initial map. The sequence of MelB$_{St}$ D59C was mutated back to wild type in COOT (*Emsley and Cowtan, 2004*). All four chains were changed to poly-Ala peptides and the side chains of each residue were rebuilt. Structure refinement was performed using phenix.real_space_refine application in PHENIX (*Afonine et al., 2013*) in real space with secondary structure and geometry restraints.

The model was refined against the final map at a resolution of 3.29 Å after sharpening using PHENIX auto_sharpen program default setting (*Terwilliger et al., 2018*). A cation Na$^+$ was modeled in the density between helices II and IV in COOT. After runs of refinement and remodeling, in total, 417 residues of MelB$_{St}$ (positions 2–210, 219–355, and 364–432), with 6 unassigned side-chains at the C-terminal domain (Leu293, Tyr355, Arg363, Tyr369, Tyr396, and Met410) due to the map disorder, 122 residues of Nb725_4 (2–123), 229 residues of NabFab H-chain (1–214), and 210 residues of NabFab L-chain (4–213), were modeled, respectively. There are three MelB$_{St}$ regions with missing densities including the middle loop righter after the Nb binding site (positions 211–218), loop$_{10-11}$ (356–361), and the C-terminal tail after residue Tyr432. In addition, the densities between positions 219–230 and the regions close to the missing densities at loop$_{10-11}$ are very weak. There is no unexplained density that can be clearly modeled by lipids. Statistics of the map reconstruction and model refinement can be found in *Table 3*. The model-map fitting quality was evaluated by Q score program (*Pintilie et al., 2020*). Pymol (*Schrodinger, 2013*) and UCSF chimera or ChimeraX (*Goddard et al., 2018*) were used to prepare the figures.

## All-atom molecular dynamics (MD) simulations

All-atom MD simulations were performed taking the inward-facing cryoEM structure as the starting model. Atom coordinates in the loops missing from the cryoEM structure were filled in using homology modeling with the Modeller software package (*Webb and Sali, 2016*). Subsequently, the protein was embedded into a lipid bilayer composed of 1-palmitoyl-2-oleoyl phosphatidylethanolamine (POPE), 1-palmitoyl-2-oleoyl phosphatidylglycerol (POPG), and cardiolipin (CDL), in a molar ratio of 7:2:1 for POPE:POPG:CDL, mimicking the membrane composition of *E. coli*. Each side of the lipid bilayer was then enclosed by a box of water molecules with 25 Å thickness. Sodium chloride ions were introduced among these water molecules to yield a solution with an approximate concentration of 0.16 M. The system setup was performed via the CHARMMGUI (*Jo et al., 2008*) web interface. The all-atom MD simulations for the outward-facing MelB$_{St}$ under the same setup have been reported by constructing through an in silico mutation, converting the outward-facing crystal structure of the D59C MelB$_{St}$ mutant (PDB ID 7L16) back to the WT MelB$_{St}$ (*Katsube et al., 2022*). In both the inward- and outward-facing configurations, a sodium ion was initially situated in the binding site, while melibiose was notably absent from the system.

Each system underwent initial optimization where the heavy atoms of proteins and lipids were harmonically restrained, employing a force constant of 1000 kJ/mol/Å$^2$ . This was followed by 100 ps of dynamics at a temperature of 300 K in the constant NVT ensemble. Subsequently, the restraints were gradually reduced during 10 ns of dynamics in the constant NPT ensemble, maintained at a temperature of 300 K and pressure of 1 atm. For equilibration, we carried out an additional 30 ns of dynamics in the NPT ensemble under the same conditions, but without any restraints. After the equilibration stage, 1 μs of dynamics was performed in the constant NPT ensemble, under the same temperature and pressure, to generate the production trajectories. We maintained the temperature using a Langevin thermostat with a friction coefficient of 1 ps$^{-1}$. The trajectory was propagated with a timestep of 2 fs, and all bonds involving a hydrogen atom were constrained. The CHARMM36 force field (*Klauda et al., 2010*; *Best et al., 2012*; *Guvench et al., 2009*) was utilized to simulate the protein, lipid, and ions throughout all simulations. Electrostatic interactions were calculated using the particle mesh Ewald method, with a tolerance of $5 \times 10^{-4}$. We set the cutoff for van der Waals interactions at 12 Å. The TIP3P model (*Jorgensen et al., 1983*) was used to treat water molecules and ions. All classical MD simulations were performed using the OPENMM (*Eastman et al., 2013*) software package.

## Hydrogen-deuterium exchange coupled to mass spectrometry (HDX-MS)

In-solution pairwise HDX-MS experiment was performed to study the structural dynamics of MelB$_{St}$. Labeling, quenching, lipids removal, and online digestion were achieved using a fully automated manner using HDx3 extended parallel system (LEAP Technologies, Morrisville, NC) (*Hamuro et al., 2003*; *Hamuro and Coales, 2018*).

Working samples of MelB$_{St}$ and MelB$_{St}$ complexed with Nb725_4 were prepared to a concentration of 50.0 μM for MelB$_{St}$ and 100 μM for Nb725_4 in a buffer of 25 mM Tris–HCl, pH 7.5, 150 mM NaCl, 10% glycerol, 0.01% DDM in H$_2$O. Aliquots of 4 μl of each working sample were diluted by tenfold into the labeling buffer of 25 mM Tris–HCl, pD 7.5, 150 mM NaCl, 10% glycerol, 0.01% DDM in D$_2$O. The reactions were incubated in D$_2$O buffer at 20°C for multiple time points (30 s, 300 s, and 3000 s) in triplicates and non-deuterated controls were prepared in a similar manner except H$_2$O buffer was used in the labeling step. The pH of the labeling buffer was measured with a pH meter mounted with a glass electrode and then corrected to pD (pD = pH + 0.4).

The reactions were quenched at given time points by adding the same volume of ice-cold 6 M urea, 100 mM citric acid, pH, 2.3 in water for 180 s at 0°C and immediately subjected to a lipid filtration module containing ZrO$_2$ available on the LEAP PAL system. After incubation of 60 s, the LEAP X-Press then compressed the filter assembly to separate proteins from the ZrO$_2$ particles-bound phospholipids and detergents. The filtered protein sample was injected into a cool box for online digestion and separation.

LC/MS bottom-up HDX was performed using a Thermo Scientific Ultimate 3000 UHPLC system and Thermo Scientific Orbitrap Eclipse Tribrid mass spectrometer. Samples were digested with a Nepenthesin-2 (Affipro, Czech Republic) column at 8°C for 180 s and then trapped in a 1.0 mm × 5.0 mm,

5.0 µm trap cartridge for desalting. Peptides were then separated on a Thermo Scientific Hypersil Gold, 50 × 1 mm, 1.9 µm, C18 column with a gradient of 10–40% gradient (A: water, 0.1% formic acid; B: acetonitrile, 0.1% formic acid) gradient for 15 min at a flow rate of 40 µL/min. To limit carry-over issues, a pepsin wash was added in between runs. To limit the back-exchange of hydrogen, all of the quenching, trapping, and separation steps were performed at near 0°C.

For data analysis, a nonspecific digested peptide database was created for $MelB_{St}$ with separate tandem mass spectrometry measurements of non-deuterated samples. Digested peptides from undeuterated $MelB_{St}$ protein were identified on the orbitrap mass spectrometer using the same LC gradient as the HDX-MS experiment with a combination of data-dependent and targeted HCD-MS2 acquisition. Using the Thermo BioPharma Finder software (v 5.1), MS2 spectra were matched to the $MelB_{St}$ sequence with fixed modifications. Altogether 153 peptide assignments (confident HDX data cross all labeling times) were confirmed for $MelB_{St}$ samples giving 86% sequence coverage. MS data files were processed using the Sierra Analytics HDExaminer software with the $MelB_{St}$ peptide database. Following the automated HDX-MS analysis, manual curation was performed. Upon the completion of the data review, a single charge state with high-quality spectra for all replicates across all HDX labeling times was chosen to represent HDX for each peptide. Differential HDX data were tested for statistical significance using the hybrid significance testing criteria method with an in-house MATLAB script, where the HDX differences at different protein states were calculated ($\Delta D = D_{Nb725\_4\text{-}bound} - D_{Nb\text{-}free}$). D denotes deuteration, defined as the mass increase above the average mass of the undeuterated peptide. Mean HDX differences from the three replicates were assigned as significant according to the hybrid criteria based on the pooled standard deviation and Welch's $t$-test with $p<0.01$. The statistically significant differences observed at each residue (i.e., $\Delta D \geq 0.3184$ for this study) were used to map HDX consensus effects based on overlapping peptides onto the structure models. Residue-level data analysis was performed using a built-in function in BioPharma Finder software. The parameters included the number of simulations of 200, recorded solutions of 20, $Chi^2$ increase by the larger of smooth absolute of 0, smother relative of 2, differential absolute of 0, and differential relative of 2.

## Acknowledgements

We thank the $S^2C^2$ cryoEM facility sponsored by NIH Common Fund Transformative High-Resolution Cryo-Electron Microscopy Program (U24 GM129541 to Wah Chiu). We thank Drs. Wah Chiu, Htet Khant, and Grigore Pintilie for support and critical discussions; Drs. Kaspar P Locher, Anthony A Kossiakoff, Joël S Bloch, and Somnath Mukherjee for kindly providing the NabFab plasmid and detailed instructions; Drs. Eric R Geertsma and Raimund Dutzler for the FX cloning tools; Dr. Gerard Leblanc for a MelB-expressing vector and DW2 strain; Drs. Bryan Sutton and Kei Nanatani for support; and Dr. Michael Wiener for critical reading and editing. This work was supported by the National Institutes of Health Grant R01 GM122759 to LG and the Welch Foundation D-2108-20220331 to RL.

## Additional information

### Competing interests

Yuqi Shi, Rosa Viner: employee of Thermo Fisher Scientific. The other authors declare that no competing interests exist.

### Funding

| Funder | Grant reference number | Author |
| --- | --- | --- |
| National Institute of General Medical Sciences | R01 GM122759 | Lan Guan |
| Welch Foundation | D-2108-20220331 | Ruibin Liang |

The funders had no role in study design, data collection and interpretation, or the decision to submit the work for publication.

## Author contributions
Parameswaran Hariharan, Yuqi Shi, Data curation, Formal analysis, Validation, Investigation, Methodology, Writing – review and editing; Satoshi Katsube, Data curation, Formal analysis, Validation, Investigation, Visualization, Methodology; Katleen Willibal, Data curation, Formal analysis, Investigation, Methodology; Nathan D Burrows, Investigation, Methodology, Writing – review and editing; Patrick Mitchell, Investigation, Methodology; Amirhossein Bakhtiiari, Samantha Stanfield, Investigation; Els Pardon, Resources, Data curation, Formal analysis, Investigation, Methodology; H Ronald Kaback, Resources; Ruibin Liang, Conceptualization, Data curation, Formal analysis, Investigation, Visualization, Writing – original draft; Jan Steyaert, Conceptualization, Resources, Supervision; Rosa Viner, Conceptualization, Resources, Supervision, Validation, Writing – original draft; Lan Guan, Conceptualization, Resources, Data curation, Formal analysis, Supervision, Funding acquisition, Validation, Investigation, Visualization, Methodology, Writing – original draft, Project administration, Writing – review and editing

## Author ORCIDs
Parameswaran Hariharan ⓘ https://orcid.org/0000-0002-6020-1547
Yuqi Shi ⓘ https://orcid.org/0009-0000-0825-7933
Nathan D Burrows ⓘ https://orcid.org/0000-0002-3973-1017
Patrick Mitchell ⓘ http://orcid.org/0000-0002-3458-6930
Els Pardon ⓘ http://orcid.org/0000-0002-2466-0172
H Ronald Kaback ⓘ https://orcid.org/0000-0003-0683-5810
Ruibin Liang ⓘ http://orcid.org/0000-0001-8741-1520
Jan Steyaert ⓘ http://orcid.org/0000-0002-3825-874X
Lan Guan ⓘ https://orcid.org/0000-0002-2274-361X

Reviewer #1 (Public Review): https://doi.org/10.7554/eLife.92462.3.sa1
Reviewer #3 (Public Review): https://doi.org/10.7554/eLife.92462.3.sa2
Author Response https://doi.org/10.7554/eLife.92462.3.sa3

# Additional files

## Supplementary files
• MDAR checklist

## Data availability
CryoEM data have been deposited to wwPDB under the accession code 8T60 and EMDB ID EMD-41062. Source data has been provided for *Figure 1*, *Figure 1—figure supplement 1*, *Figure 2—figure supplement 8*, and *Figures 5 and 6*.

The following datasets were generated:

| Author(s) | Year | Dataset title | Dataset URL | Database and Identifier |
|---|---|---|---|---|
| Guan L | 2024 | Mobile barrier mechanisms for Na+-coupled symport in an MFS sugar transporter | https://www.rcsb.org/structure/8T60 | RCSB Protein Data Bank, 8T60 |
| Guan L | 2024 | Mobile barrier mechanisms for Na+-coupled symport in an MFS sugar transporter | https://www.ebi.ac.uk/emdb/EMD-41062 | Electron Microscopy Data Bank, EMD-41062 |

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
