## [Editor Report · eLife assessment]

In an **important** study that will be of interest to the mechanistic membrane transport community, the authors capture the first cryoEM structure of the inward-facing melibiose transporter MelB, a well-studied model transporter from the major facilitator (MFS) superfamily. CryoEM experiments and supporting biophysical experiments provide **solid** evidence for transporter conformational changes.

---

## [Referee Report · Reviewer #1 (Public Review)]

Summary:

The current study reports a cryo-EM structure of MFS transporter MelB trapped in an inward-facing state by a conformationally selective nanobody. The authors compare this structure to previously-resolved crystal structures of outward-facing MelB. Additionally, the authors report H/D exchange/ mass spec experiments that identify accessible residues in the protein.

Strengths:

The authors overcame very significant technical challenges to solve the first inward-facing structure of the small, model MFS transporter MelB by cryo-EM. The use of conformation-trapping nanobodies (which had been reported previously by this group) is particularly nice.

Weaknesses:

The authors highlight the use of HDX experiments as a measurement of protein conformational dynamics. However, the experiment instead measures the accessibility of different residues. An ideal experiment would trap the transporter in inward- and outward states, but only the inward conformation is trapped here. The outward-facing conformation is instead an ensemble of outward and occluded conformations. It seems obvious that this will be more dynamic than the nanobody-trapped inward state.

---

## [Referee Report · Reviewer #3 (Public Review)]

Summary:

The manuscript authored by Lan Guan and colleagues reveals the structure of the cytosol-facing conformation of the MelB sodium/Li coupled permease using the nab-Fab approach and cryoEM for structure determination. The study reveals the conformational transitions in the melB transport cycle and allows understanding of the role of sugar and ion specificities within this transporter.

Strengths:

The study employs a very exciting strategy of transferring the CDRS of a conformation specific nano body to the nab-fab system to determine the inward-open structure of MelB. The resolution of the structure is reasonable enough to support the major conclusions of the study. This is a well-executed study.

---

## [Author Response]

The following is the authors’ response to the original reviews.

**Reviewer #1 (Recommendations For The Authors):**
It is somewhat speculative that the structure represents the EIIa-bound regulatory state. There's a strong enough case that it should be analyzed in the discussion, but I don't think it is firmly established. Therefore, the title of the paper should be changed.

Our answer: Thank you for the comment. We have changed the title to “Mobile barrier mechanisms for Na+-coupled symport in an MFS sugar transporter”

Reading through the manuscript, it was challenging to distinguish what is new in the current manuscript and what has been done previously. There were a lot of parts where it was hard for me to identify the main point of the current study among all the details of previous studies. It would also benefit from shortening. For example:-Page 6: Nb725 binding has already been characterized extensively in the very nice JBC paper earlier this year. It's important to test 725-4 for binding, but since it doesn't change the binding interaction, and probably wouldn't be expected to, the entire section could be written more succinctly. The main point, which is that 725-4 behaves like 725, is lost among all the details

Our answer: Thanks for this instructive suggestion. We have shortened the description in this section.

-Page 9-10. I don't understand what summarizing all of the results from the previous D59C studies adds to the current story. It's important because it provides an indication of the substrate binding site, but its mechanism of action does not seem relevant to the current work.

Our answer: We have shortened the description of the sugar-binding site and moved the previous Fig. 3b to supplementary figure sFig. 11. According to your comment about showing the location of the binding sites, which is also suggested by Reviewer #2, we modified Fig. 3 and added two panels to map the location of the bound Na+ in the inward-facing structure and the bound sugar in the outward-facing structure.

The sugar-binding site identified in the published structure is critical to construct the mobile barrier mechanism. The sugar-binding residues identified in the published structure provided essential data to support the conclusion that the sugar-binding pocket is broken in the inward-facing structure. Thus, this published structure is mechanistically relevant to the current study.

-Page 12. Too much summary of the previous outward structure. Since this is already part of the literature, it would be more efficient to reference the previous data when it is important to interpret the new data (or show as a figure).

Our answer: The introduction of the previous sugar-binding sit is important for the detailed comparison between the two states as discussed above, but we agree with this reviewer and have significantly shortened the paragraph by moving the detailed description into the legend to the sFig. 11.

-Instead of providing the PDB ID in figures of the current structure, just say "current work" or similar. Then it is obvious you are not citing a previous structure.

Our answer: To distinguish clearly the new data and published results, the citation of the cryoEM structure [PDP ID 8T60] has been completely removed from the main text but kept in sTable 1.

-An entire panel of Figure 3 is dedicated to ligand binding in a previous outward-facing structure.Showing it in the overlay would be sufficient.

Our answer: It is the first time for us to show a structure with a bound-Na+. Fig. 3 also illustrates the spatial relationship between the sugar-binding pocket and the cation-binding pocket since both binding sites are determined now. As stated above, according to two reviewers’ comments, we have modified the Figures and the Fig. 3d is the overlay.

Please increase the size of the font in all figures. It should be 6-8 point when printed on a standard sheet of paper. Labels in Figure 3, distances in Figure 4, and everything in Figure 5 is hard to see.

Our answer: Thank you for the comments and the enlargement of the figure size and label font in all figures have been made.

Figure 2: would be helpful to show Figure S8 in the main text, orienting the reader to the approximate location of substrate binding. What is known about the EIIA-Glc binding interface? Has anyone probed this by mutagenesis? Where are these residues on the overall structure, and are they somewhere other than the nanobody interface?

Our answer: Thank you for this comment. We have added a panel for orienting the readers about the substrate location in MelB in Figure 3c. The sFig. 8 actually focuses on the details of Nb interactions with MelB. Our current data strongly supported the notion that the Nb-bound MelBSt structure mimics the EIIAGlc-bound MelB but is not structurally resolved, so we have tuned down our statement on EIIAGlc. There is one study suggesting the C-terminal tail helix may be involved in the EIIAGlc binding, which has been added to the discussion.

Can Figure 5 be split into 2 figures and simplified?

Our answer: thanks for the suggestion. We have split it into Figs. 5b and 6 and also moved the peptide mapping to the Fig 5a.

What is the difference between cartoon and ribbon rendering?

Our answer: Ribbon: illustrating the structure; cartoon: highlighting the positions with statistically significant protection or deprotection. The statistically significant changes are implied by the ribbon representation; Sphere: not covered by labeled peptides.

Can the panels showing the kinetic data be enlarged? I don't think they need to surround the molecule. An array underneath would be fine.

Our answer: We have enlarged all figures and labels. The placement of selected plots around the model could clearly show the difference in deuterium uptake rates between the transmembrane domain and extra-membrane regions. We will maintain this arrangement.

Do colors in panel A correspond with colors in panel B?

Our answer: The color usage in both are different. Now the two panels have been separated.

Do I understand correctly that in the HDX experiments, negative values indicate positions that exchange more quickly in the nanobody-free protein relative to the nanobody-bound protein?

Our answer: Your understanding is correct.

I assume some of this is due to the protein changing conformation, but some of it might be due to burial at the nanobody-binding interface. Can those peptides be indicated?

Our answer: Thank you for this comment. We have marked the peptide carrying the Nb-binding residues on uptake plots in Figs.6 and Extended Fig. 1. There are only three Nb-binding residues covered by many overlapping peptides. Most are not covered, either not carried by the labeled peptides (Tyr205, Ser206, and Ser207) or with insignificant changes (Pro132 and Thr133), except for Asp137, Lys138, and Arg141 which are presented in 8 labeled peptides.

Few buried positions in the outward-facing state are expected to be solvent in the inward-facing state; unfortunately, inward-facing state they are buried by Nb binding.

Make figure legends easier to interpret by removing non-essential methods details (like buffer conditions).

Our answer: We removed the detailed method descriptions in most figure legends. Thank you.

Check throughout for typos.ie page 9 Lue LeuPage 9 like likely

Our answer: We have corrected them. Thank you!

**Reviewer #2 (Recommendations For The Authors):**
I have mostly minor questions/remarks.Why not do the hdx-ms experiments in the presence of sugar? That would give a proper distinction between two conformational states, instead of an ensemble of states vs one state.

Our answer: MelB conformation induced by sugar is also multiple states, and likely most are outward-facing states and occluded intermediate states. This is also supported by the new finding of an inward state with low sugar affinity. The ideal design should be one inward and one outward to understand the inward-outward transition. We have not identified an outward-facing mutant while we can obtain the inward by the Nb. WT MelBSt with bound Na+ favors the outward-facing state. Although our design is not ideal, we do have one state vs a predominant outward-facing WT with bound Na+.

Minor comments:• Fig 5 is misleading as the peptide number does not match with the amino acid sequence. I would suggest putting a heat map with coverage on top. Or showing deuterium uptake per peptide. See examples below.

Our answer: The peptide number should not match with sequence number. We have 155 overlapping peptides that cover the entire amino acid sequence including the 10-His tag, and there are 60 residues with no data because they are not covered by a labeled peptide. The residue positions that are covered by peptides are estimated by bars on the top. The cylinder length does not correspond to the length of the transmembrane helix, just for mapping purposes.

Can the authors explain how they found that the Nbs bind to the cytoplasmic side (before obtaining the structure)?

Our answer: Our in vivo two-hybrid assay between the Nb and MelBSt indicated their interaction on the cytoplasmic surface of MelBSt, which is further confirmed by the melibiose fermentation and transport assay, where the transport activities were completely inhibited by intracellularly coexpressed Nb and MelBSt. Thanks for raising this question.

• The authors use the word "substrate" indifferently for sugar and Na+ binding, which is a bit confusing. Technically, only sugar is the substrate and Na+ is a ligand, or cotransported-ion, that powers the reaction of transport. This might sound like nit-picking but it can lead to misunderstandings (at some point I thought two sugars were transported, and then I was looking for the second Na+ binding site).

Our answer: We used to call the sugar and Na as co-substrate but we agree with this comment.

We have changed by using substrate for the cargo sugar and coupling cation for the driving cation.

• Abstract "only the inner barrier" - the is missing.

Thanks. We have corrected this.

• p.3 intro "and identified that the positive cooperativity of cation and melibiose, " something is missing.

Thanks again. We missed the “as the core symport mechanism”.

• P.6 Nb275_4 instead of Nb725_4

Thank you very much for your careful reading.

• P.7. Also, affinity affinities

Thank you very much. We changed to “; and also, the α-NPG affinity decreased by 21~32-fold for both Nbs”

• P.8 " contains 417 MelBSt residues (positions 2-210, 219-355, and 364-432). This does not sum up to 417 residues.

Thanks for your critical reading. We changed 364-432 to 262-432.

• p.9 Lue 54

We have corrected it to Leu54.

• I find fig.3 hard to read. Can the authors show the Na+ binding pockets and sugar binding pockets within the structure? Especially figure 3b. why are the residues in different colors?

Our answer: We have moved Fig 3b into sFig. 11. We colored the residues in the previous Fig 3B to match the hosting helices. We have added two panels to show the location of both sugar and Na in the molecular. Thank you for your comments.

• Fig4 bcef. Colored circles at the end of the helices. What are they for?

Our answer: We revised the legend. “The paired helices involved in either barrier formation were highlighted in the same colored circles.”

• 86% coverage includes the his-tag - it would be good to clarify that.

Our answer: Yes, it includes the 10-His tag.

• Fig.7 - anti clockwise cycle of transport is counter-intuitive.

Our answer: We have re-arranged. Our model was constructed originally to explain efflux due to limited information at the earlier state. Now more data are available allowing us to explain inflow and active transport.

• Where are all the uptake plots per peptide for the HDX-MS data?

Our answer: We have added the course raw data and prepared all uptake plots for all 71 peptides with statistically significant changes as an Extended Fig. 1.

• P.22 protein was concentrated to 50 mg/mL. Really? That is a lot.

This is correct. We can even concentrate MelBSt protein to greater than 50 mg/ml.

• Have the authors looked into the potential role of lipids in regulating the conformational transition? Since the structure was obtained in nanodiscs, have they observed some unexplained densities? The role of lipid-protein interactions in regulating such transitions was observed for several transporters including MFS (Gupta K, et al. The role of interfacial lipids in stabilizing membrane protein oligomers. Nature. 2017 10.1038/nature20820. Martens C, et al. Direct protein-lipid interactions shape the conformational landscape of secondary transporters. Nat Commun. 2018 10.1038/s41467-018-06704-1.). Furthermore, I see the authors have already observed lipid specific functional regulation of MelB (ref: Hariharan, P., et al BMC Biol 16, 85 (2018). https://doi.org/10.1186/s12915-018-0553-0). A few words about this previous work, and even commenting on the absence of lipid-protein interactions in this current work is worthwhile.

Our answer: Thanks for this very relevant comment. We paid attention to the unmodelled densities. There is one with potential but it is challenging to model it. We have added a sentence “There is no unexplained density that can be clearly modeled by lipids.” in the method to address this concern.

**Reviewer #3 (Recommendations For The Authors):**
1. In the following sentence, the authors report high errors for the Kd value. The anti-Fab Nb binding to NabFab was two-fold poorer than Nb725_4 at a Kd value of 0.11 {plus minus} 0.16 μM. The figure however indicates that the error value is 0.016 µM. Pls correct.

Our answer: Thank you. You are correct. The error has been corrected. 0.16 ± 0.02 uM. In this revised manuscript, we present the data in nM units.

1. Is the stoichiometry of the MelB:Na+ symport clearly known in this transporter. It can be mentioned in the discussion with appropriate references.

Our answer: Yes, the stoichiometry of unity has been clearly determined, which was included in the second paragraph of the previous version.

1. In the last section of results, the authors seem to suggest a greater movement within their Cterminal helical bundle compared to N-terminal helices. Is there evidence to suggest an asymmetry in the rocker switch between the two states of the transporter?

Our answer: Our structural data revealed that the C-terminal bundle is more dynamic compared with the N-terminal bundle where hosts the residues for specific binding of galactoside and Na+. The HDX data showed that the most dynamic regions are the structurally unresolved C-terminal tail by either method, the conserved tail helix and the middle-loop helix. transmembrane helices are relatively less dynamic with similar distributions on both transmembrane bundles. Since the most dynamic regions are peripheral element associated with the C-terminal domain, it might give a wrong impression. With regard to the symmetric or asymmetric movement, which will certainly affect the dynamic interactions between the transporter and the lipids, we favor the notion that MelBSt performs symmetric movement during the rocker switch between inward and outward states at the least cost for the protein-lipids interaction.

1. Figure 1. Are the thermograms exothermic or endothermic? clarify

Our answer: In our thermograms, all positive peaks are exothermic due to the direct detection of the heat release by the TA instrument. We clarified this in Method and now we stress this in figure legends to avoid confusion.

1. Figure 4a,d. Please put in a membrane bilayer and depict cytosolic and extracellular compartments for clarity.

Thank you. We have added a bilayer and labeled the sidedness in this figure and other related figures.

1. Fig 7. Melibiose symport cannot be referred to as Melibiose efflux transport in the legend as the latter refers to antiport. Pls rectify.

Our answer: Influx and efflux are conventionally used to describe the direction of movement of a substrate. The use of symport and antiport indicates the directions of the coupling reaction for the cargo and cation. For the symporter MelB, melibiose efflux means that sugar with the coupled cation moves out, which is driven by the melibiose concentration. During the steady state of melibiose active transport, efflux rate = influx rate.

1. Page 11 "A common feature of carrier transporters". The authors can use either carriers or transporters. Need not use both simultaneously.

Sorry for overlooking this. We have deleted carriers. Thank you very much for your time.

1. Several typos were noticed in this manuscript. some are listed below. pls correct.Page 4- last paragraph "Furthermore"

We have corrected it. Thank you again!

Page 7 - second para one repharse "affinity reduced by 21~32 fold/units.." pls clarify

Added 21~32 fold.

Page 9 - "so it is highly likely that inward-open conformation" pls correct.

We have corrected to “likely”.

Fig. S9c - correct the spelling "Distance".

We have corrected to “Distance”